# Employing airborne radiation and cloud microphysics observations to improve cloud representation in ICON at kilometer-scale resolution in the Arctic

Jan Kretzschmar[1], Johannes Stapf[1], Daniel Klocke[2,3], Manfred Wendisch[1], and Johannes Quaas[1]

[1]Institute for Meteorology, Universität Leipzig, Leipzig, Germany
[2]Deutscher Wetterdienst, Offenbach, Germany
[3]Hans-Ertel-Zentrum für Wetterforschung, Offenbach, Germany

**Correspondence:** Jan Kretzschmar (jan.kretzschmar@uni-leipzig.de)

**Abstract.** Clouds play a potentially important role in Arctic climate change, but are poorly represented in current atmospheric models across scales. To improve the representation of Arctic clouds in models, it is necessary to compare models to observations to consequently reduce this uncertainty. This study compares aircraft observations from the Arctic Cloud Observations Using Airborne Measurements during Polar Day (ACLOUD) campaign around Svalbard, Norway in May/June 2017 and simulations using the ICON (ICOsahedral Non-hydrostatic) model in its numerical weather prediction (NWP) set-up at 1.2 km horizontal resolution. By comparing measurements of solar and terrestrial irradiances during ACLOUD flights to the respective properties in ICON, we showed that the model systematically overestimates the transmissivity of the mostly liquid clouds during the campaign. This model bias is traced back to the way cloud condensation nuclei (CCN) get activated into cloud droplets in the two-moment, bulk microphysical scheme used in this study. This process is parameterized as function of grid-scale vertical velocity in the microphysical scheme used, but in-cloud turbulence cannot sufficiently be resolved at 1.2 km horizontal resolution in Arctic clouds. By parameterizing subgrid-scale vertical motion as a function of turbulent kinetic energy, we are able to achieve a more realistic CCN activation into cloud droplets. Additionally, we showed that by scaling the presently used CCN activation profile, the hydrometeor number concentration could be modified to be in better agreement with ACLOUD observations in our revised CCN activation parameterization. This consequently results in an improved representation of cloud optical properties in our ICON simulations.

## 1 Introduction

In recent decades, the Arctic has proven to be especially susceptible to global climate change (Screen and Simmonds, 2010), as several positive feedback mechanisms strengthen the warming in high latitudes of the Northern Hemisphere (Serreze and Barry, 2011; Wendisch et al., 2017). Among those feedback mechanisms that influence the Arctic climate, the cloud feedback - even

though being small in magnitude compared to other feedback mechanisms like the surface albedo or temperature feedbacks - exhibits a relatively large uncertainty (Pithan and Mauritsen, 2014; Block et al., 2020). This uncertainty can be related to the general complexity of the Arctic climate system and to misrepresented microphysical processes in global climate models (GCMs) that are used to quantify the cloud feedback. Typical issues associated with the simulation of clouds in the Arctic are

incorrectly simulated amount and distribution of clouds (English et al., 2015; Boeke and Taylor, 2016), which often can be linked to an erroneous representation of mixed-phase clouds (Cesana et al., 2012; Pithan et al., 2014; Kretzschmar et al., 2019). This consequently affects the quantification of the effect of Arctic clouds on the (surface) energy budget in GCMs (Karlsson and Svensson, 2013).

To identify processes within the microphysical parametrization that are misrepresented in models, it is inevitable to compare
them to appropriate observations (Lohmann et al., 2007). As pointed out by Kay et al. (2016), any comparison between modeled and observed quantities can easily be misleading if it is not scale- and definition-aware. For GCMs, observations from satellite remote sensing are well suited, being on similar scales as those large scale models. A comparison to satellite-derived quantities can further be made definition-aware by using instrument simulators like they are provided within the Cloud Feedback Model Intercomparison Project's (CFMIP) Observation Simulator Package (COSP; Bodas-Salcedo et al., 2011). The benefit of using
COSP for evaluating clouds in GCMs in the Arctic has been shown in several studies (Barton et al., 2012; Kay et al., 2016; Kretzschmar et al., 2019).

Even though satellite observations provide valuable information on the atmospheric state in the Arctic, they often suffer from instrument-dependent idiosyncrasies like ground clutter for a space-borne cloud radar or attenuation of the beam of a space-borne lidar by optically thick clouds (Cesana et al., 2012). Those problems can be in part overcome by using ground-based or
aircraft observations. Due to much smaller temporal and spatial scales, those observations only have limited suitability for the evaluation of large-scale models. To this end, the use of storm-resolving models with grid sizes on the order of kilometers or large eddy models is necessary, as they are able to better capture features and variability present in those rather smaller scale observations (Stevens et al., 2019). Due to the relatively large computational effort that is needed for large eddy simulations, they are limited in spatial extent and are often used for comparison with ground based observations at individual locations in
the Arctic (e.g. Loewe et al., 2017; Sotiropoulou et al., 2018; Neggers et al., 2019; Schemann and Ebell, 2020). Furthermore, large eddy simulation have been used to study and evaluate microphysical processes (e.g. Fridlind et al., 2007; Ovchinnikov et al., 2014; Solomon et al., 2015), as well as aerosol-cloud interactions (e.g. Possner et al., 2017; Solomon et al., 2018; Eirund et al., 2019) in the Arctic. To avoid the need for large computational resources but still be able to resolve many processes that act on scales that cannot be captured by GCMs, limited area simulations with grid sizes on the order of a few kilometers, where
(deep) convection does not need to be explicitly parameterized, can offer a good compromise. Simulations at such resolutions on relatively large domains have received increased interest in recent years (Stevens et al., 2019).

This study makes use of such a set-up using the ICOsahedral Non-hydrostatic (ICON) model (Zängl et al., 2015) at kilometer-scale horizontal resolution. Studies, mainly focusing on the tropical Atlantic, have reported that the model at storm-resolving resolutions is able to simulate the basic structure of clouds and precipitation in that region (Klocke et al., 2017; Stevens et al.,
2020). In the present study, ICON is used in a similar set-up and is compared to observations that have been derived from the

Arctic Cloud Observations Using Airborne Measurements during Polar Day (ACLOUD) campaign around Svalbard, Norway (Wendisch et al., 2019; Ehrlich et al., 2019) and to observations derived during the Physical Feedbacks of Arctic Boundary Layer, Sea Ice, Cloud and Aerosol (PASCAL; Flores and Macke, 2018) ship-borne observational campaign in the sea ice covered ocean north of Svalbard in May and June 2017. This study mainly compares observations of solar and terrestrial irradiances during ACLOUD flights to our ICON simulations to obtain a first estimate whether the model is able to correctly simulate general cloud optical properties. Based on the results of this comparison, it is further explored to what extent cloud macro- and microphysical properties might be misrepresented in this set-up and how to improve the simulation of clouds in ICON at kilometer-scale.

## 2  Data and model

### 2.1  ACLOUD/PASCAL campaign

In May and June 2017, two concerted field studies took place around Svalbard, Norway (Wendisch et al., 2019): the Arctic Cloud Observations Using Airborne Measurements during Polar Day (ACLOUD; Ehrlich et al., 2019) campaign and the Physical Feedbacks of Arctic Boundary Layer, Sea Ice, Cloud and Aerosol (PASCAL; Flores and Macke, 2018) ship-borne observational study. The airborne measurements during ACLOUD where conducted with the two research aircraft Polar 5 and Polar 6 (Wesche et al., 2016) that were based in Longyearbyen (LYR), Norway. While Polar 5 focused on remote sensing observations of mainly low-level clouds and surface properties from higher altitudes (2-4 km), Polar 6 concentrated on in situ observations of cloud microphysical and aerosol properties, in and below the clouds. Ground-based observations from the ship and an ice floe in the sea-ice-covered ocean north of Svalbard were performed during PASCAL using the German research vessel (R/V) Polarstern (Knust, 2017). Additionally, a tethered balloon was operated on an ice floe camp during PASCAL (Egerer et al., 2019).

The synoptic development during both campaigns is separated into three phases (Knudsen et al., 2018). A period with advection of cold and dry air from the north in the beginning (23-29 May 2017) was followed by a warm and moist air intrusion into the region where the two campaigns took place (30 May -12 June 2017). During the final two weeks of the campaigns (13-26 June 2017), a mixture of warm and cold air masses prevailed. Especially during the last two phases, clouds in the domain close to Polarstern, where the bulk of the measurements took place, mainly consisted of (super-cooled) liquid clouds with only little cloud ice being present (Wendisch et al., 2019).

In the following, a brief description of the instrumentation and data used in this study is given (for a comprehensive overview we refer the reader to Wendisch et al. (2019) and Ehrlich et al. (2019)). Two pairs of upward and downward looking CMP 22 pyranometers for the solar (0.2-3.6 $\mu$m) and CGR4 pyrgeometers for major parts of the terrestrial spectral range (4.5-42 $\mu$m) were installed on board of Polar 5 and Polar 6 to measure the upward and downward broadband (solar and terrestrial) irradiances on both aircraft (Stapf et al., 2019). We also utilize microphysical data that have been derived from in-situ measurements on Polar 6. We use data of the particle size number distribution obtained from the Small Ice Detector Mark 3 (SID-3) (Schnaiter and Järvinen, 2019) covering a size range of 5-45 $\mu$m divided into 16 size bins (2-5 $\mu$m resolution). For more information on

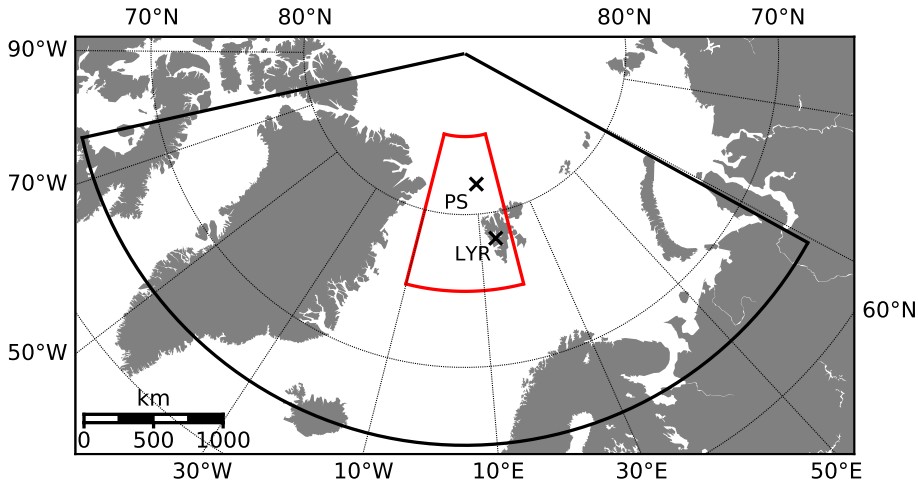

**Figure 1.** Set-up of the limited-area simulations. The outer domain (black) has an approximate resolution of 2.4 km, while the inner domain (red) has a resolution of 1.2 km. Additionally marked is Longyearbyen/Norway (LYR) where Polar 5 and Polar 6 were stationed during ACLOUD, as well as the postion of the R/V Polarstern (PS) during the ice floe camp.

the SID-3 and processing of the measurements, the reader is referred to Schnaiter et al. (2016) and Ehrlich et al. (2019). For comparison of the bulk liquid water content, we exploit data from a Nevzorov probe (Korolev et al., 1998) that was installed on Polar 6 (Chechin, 2019). Furthermore, we use observations of cloud base height as observed by the laser ceilometer and cloud top height derived from a 35 GHz cloud radar (Griesche et al., 2019) onboard R/V Polarstern to derive geometrical cloud depth in the sea ice-covered ocean north of Svalbard.

## 2.2 ICON simulations

In this study, data measured during ACLOUD and PASCAL are compared to the output of the ICOsahedral Non-hydrostatic model (ICON; Zängl et al., 2015). ICON is a unified modelling systems that allows simulations on several spatial and temporal scales, spanning from simulation of the global climate on the one end (Giorgetta et al., 2018) to high resolution large eddy simulations (LES) on the other (Dipankar et al., 2015; Heinze et al., 2017). ICON is also employed as a numerical weather prediction (NWP) model at the German Meteorological Service (Deutscher Wetterdienst, DWD). For each application (GCM, NWP, LES), a dedicated package of physical parametrizations is provided to satisfy the specific needs for each set-up. For our simulations, the applied set of physical parametrizations is similar to that used in Klocke et al. (2017). However, we use the two-moment, bulk microphysical scheme developed by Seifert and Beheng (2006) instead of the single moment scheme by Baldauf et al. (2011) used in Klocke et al. (2017). Furthermore, we apply an all-or-nothing cloud cover scheme that allows for grid-scale clouds only as this facilitates the comparison with the observations. At the resolutions used in this study, an all-or-nothing cloud cover scheme might miss some clouds as the necessary saturation humidity might not be reached. A comparison to simulations with a fractional cloud cover scheme showed only little differences compared to the all-or-nothing cloud cover

scheme used, which made us confident that resolving clouds at grid scale only is sufficient for our set-up. The Rapid Radiation Transfer Model (RRTM; Mlawer et al., 1997) is applied to derive the radiative fluxes. Due to the rather fine horizontal resolution of our simulations, we only parametrized shallow convection using the Tiedtke (1989) shallow convection parameterization

with modifications by Bechtold et al. (2008), whereas deep convection is considered resolved (albeit not relevant for the Arctic case considered here). In the following, the used set-up will be simply denoted as ICON. However, findings in this study are specific to our chosen set-up (spatial scale and parameterizations used) and should not be seen as generally representative for ICON.

We deploy ICON in a limited-area set-up with one local refinement (nest) in the region where the research flights and ship

observations were performed (Figure 1). The outer domain has a horizontal resolution of approximately 2.4 km (R2B10 in the triangular refinement) while the inner nest has a refined resolution (R2B11) of approximately 1.2 km. For both domains, we use 75 vertical levels spanning from the surface to 30 km altitude with a vertical resolution of 20 m at the lowest model level that gradually gets coarser towards model top. We initialize the model using the analysis of European Center of Medium Weather Forecast (ECMWF) Integrated Forecasting System (IFS). The respective IFS forecast is used as boundary data to

which we nudge our model every three hours. We do not continuously run the model for the whole period of the campaign but re-initialize the model from the 1200 UTC analysis of the previous day in case of a subsequent day with flight activities. This gives the model a spin-up time of more than 12 hours even for takeoffs in the early morning.

During the initial comparison of ICON and the ACLOUD observations, we found that the albedo of sea ice in the model is substantially lower compared to values observed during ACLOUD (Wendisch et al., 2019). The reason for this underestimation

of the surface albedo in ICON is caused by how our simulations are initialized using the IFS analysis. As the IFS sea ice albedo is not used during the initialization of ICON, the parametrization of the sea ice albedo performs a cold start. For such a cold start, the sea ice albedo is a function of the sea ice surface temperature only, as given by Mironov et al. (2012) (their Equation 5). This formulation was slightly adapted in ICON by setting the maximum sea ice albedo ($\alpha_{\mathrm{max}}$) to 0.70 and the minimum sea ice albedo ($\alpha_{\mathrm{min}}$) to 0.48. For surface temperatures close the freezing point (as it has been observed during ACLOUD,

especially in the second half of the campaign), such a cold start results in albedo values that are considerably lower compared to the observations. This underestimation of the sea ice albedo could be avoided by increasing the spin-up of the model to a few weeks or by using DWD ICON analysis instead of the IFS analysis. In the latter case, the albedo is initialized from the initial data and no spin-up is required (Wendisch et al., 2019). As one of the main aims of this comparison are irradiances, an accurate representation of surface albedo is crucial and we, therefore, chose to take yet another approach. Due to the fact

that the simulated period falls into the onset of the melting period, the sea ice albedo significantly reduces in that period. To accurately represent this reduction in sea ice albedo, we prescribe the sea ice albedo as a function of time to be consistent with the observed sea ice albedo. For this purpose, from the observations, only scenes with homogeneous sea ice are selected using a fish-eye-camera-derived sea ice concentration threshold of 95 %. This approach by construction results in a standard deviation of as little as 0.024 between daily modeled and observed albedo. In case of fractional sea ice cover in the model, the surface

albedo is a surface fraction-weighted average between the prescribed value and the albedo of open water (taken as 0.07).

For the comparison of our ICON simulations to the ACLOUD data, we temporally and spatially collocate the model output to

be consistent with the actual position and altitude of the aircraft. We use a multidimensional binary search tree (also known as k-d tree; Bentley, 1975) to sample the model output along the flight track in space and time directly on its native unstructured, triangular grid. The temporal frequency of the observational data is 1 Hz. Additionally, we averaged the (sampled) datapoints from the observations and the simulations into 20 second intervals. This ensures that the observational data is on a similar spatial scale as the simulation on the 1.2 km grid of the inner domain (considering an average velocity of the aircraft of $60 \, \mathrm{m\,s^{-1}}$). Due to storage constraints, we chose to output the model state only every 30 minutes, which reduces temporal variability in the model output. As the planes are not static and "fly" through the model grid, temporal variability is, to some extent, replaced by spatial variability when sampling a large-enough area along the flight track. Additionally, the 30 minute output frequency introduces inconsistencies in the top of atmosphere incoming solar irradiance, as the solar zenith angle is constant in the model output while it varies with time in the observations. This implies that the largest temporal difference between an observational data point and the output timestep of ICON is $\pm$ 15 minutes, causing a bias of up to $\pm 14 \, \mathrm{W\,m^{-2}}$ for incoming solar irradiation at the top of the atmosphere in the early morning and late evening when the temporal derivative of incoming solar radiation is the largest. As most of flights took place during noon and we mostly focus on cloudy conditions, we expect this bias to be on the order of a few $\mathrm{W\,m^{-2}}$ at most, making us confident that this issue will not significantly influence the overall findings in this study. Even though being on similar scales, spatial and temporal variability in both datasets prohibit a one-to-one comparison. We will, therefore, use histograms in the comparison.

## 3 Surface radiative quantities as simulated with ICON and measured during ACLOUD

In the following, the simulations are compared to data for several surface radiative variables that have been observed during low-level flight sections. Some flights were excluded due to relatively short flight times to save computational resources. Additionally, some flights with cloudless conditions towards the end of the campaign were not analyzed as the main focus of this study is a comparison of cloud properties. An overview of the flights used for the comparison is given in Table 1. In the observation and in the model, we define low-level flight sections as such that no cloud is present below the present altitude of the aircraft.

### 3.1 Spatial structure of the radiative field of the Arctic atmospheric boundary layer

In the Arctic, two distinct radiative states have been reported: a radiatively clear state with no, or only radiatively thin clouds and a cloudy state with opaque clouds (Shupe and Intrieri, 2004; Stramler et al., 2011). This two-state structure was also observed during ACLOUD, but compared to spatially fixed observations with almost constant surface albedo, observations during ACLOUD were further decomposed into a cloudy and cloudless state over sea ice and open ocean, which consequently results in a four-state structure (Wendisch et al., 2019). As in Wendisch et al. (2019), we compiled two-dimensional histograms of surface albedo and surface net terrestrial and net solar irradiances, defined as the difference between downward and upward radiative energy flux densities, for the ACLOUD observations and the ICON simulations (Figure 2). The general difference to Wendisch et al. (2019) (their Figure 14) is explained by the prescribed surface albedo approach applied in this study, which

**Table 1.** Flights used for the comparison to ICON simulations (approximately 116 flight hours). The values given for the low-level scenes corresponds to the number of the averaged 20 second intervals used in the following comparison. For more information on the scientific target of each research flight, refer to Wendisch et al. (2019) and Ehrlich et al. (2019).

| Flight No. | Date in 2017 | Flight Time (UTC) | | Low-level scenes | |
|:---:|:---:|:---:|:---:|:---:|:---:|
| | | Polar 5 | Polar 6 | all-sky + all surfaces | cloudy + sea ice |
| 4 | 23 May | 09:12-14:25 | - | 69 | 12 |
| 5 | 25 May | 08:18-12:46 | - | - | - |
| 6 | 27 May | 07:58-11:26 | - | - | - |
| 7 | 27 May | 13:05-16:23 | 13:02-16:27 | 58 | - |
| 8 | 29 May | 04:54-07:51 | 05:11-09:17 | 60 | - |
| 10 | 31 May | 15:05-18:57 | 14:59-19:03 | 199 | - |
| 11 | 2 June | 08:13-13:55 | 08:27-14:09 | 73 | 7 |
| 12 | 4 June | - | 10:06-15:39 | 65 | 55 |
| 13 | 5 June | 10:48-14:59 | 10:43-14:44 | 101 | 70 |
| 14 | 8 June | 07:36-12:51 | 07:30-13:20 | 80 | 6 |
| 17 | 14 June | 12:48-18:50 | 12:54-17:37 | 275 | 275 |
| 18 | 16 June | 04:45-10:01 | 04:40-10:31 | - | - |
| 19 | 17 June | 09:55-15:25 | 10:10-15:55 | 95 | 22 |
| 20 | 18 June | 12:03-17:55 | 12:25-17:50 | 131 | - |
| 23 | 25 June | 11:09-17:11 | 11:03-16:56 | 347 | - |

results in higher sea ice albedo values compared to the previously used model set-up.

In general, the structure of the modeled net terrestrial irradiance ($F_{net,terr}$) close to the surface (Figure 2 (a) and (b)) is in agreement with the observed one. Only for surface albedo values between 0.6 and 0.7, noticeable differences between the ACLOUD observations and the ICON simulations become obvious. Those albedo values are related to days towards the end of the campaign (mid/late June 2017) when the melting season had begun and sea ice albedo was reduced. For this period, the model overestimates the presence of cloudy conditions whereas cloudless conditions were present in the ACLOUD observa-

tions. Conversely, for situations with sea ice albedo greater than 0.7, ICON overestimates the presences of cloudless conditions. The lack of cloudless conditions for surface albedo values between 0.6 and 0.7 in the ICON simulations is also visible from the histograms of surface albedo and net solar irradiance (Figure 2 (c) and (d)). For surface albedo larger than 0.7, the net

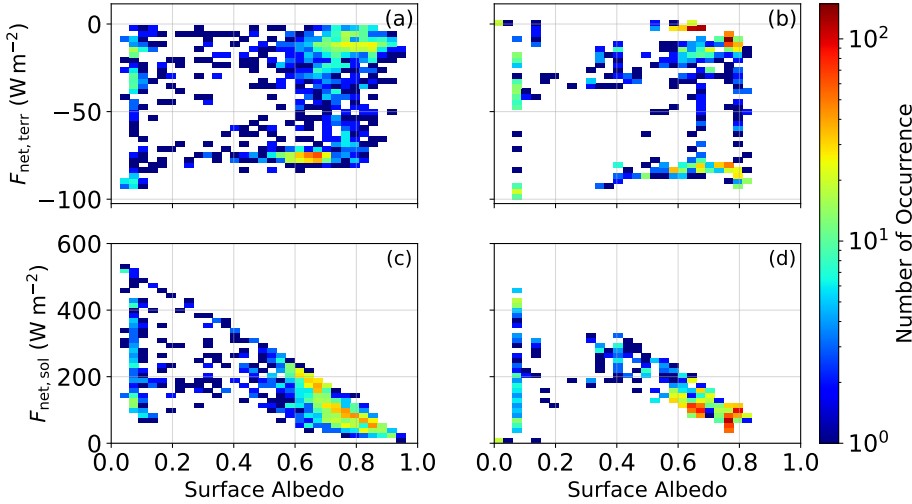

**Figure 2.** Two dimensional histograms of surface albedo and (top row; a, b) net terrestrial-/ (bottom row; c, d) net solar irradiance at the surface (W m$^{-2}$) for (left column; a, c) ACLOUD observations and (right column; b, d) ICON simulations.

solar irradiance ($F_{net,sol}$) close to the surface seems, on average, in agreement with the observations, even though the observed variability in surface albedo is not simulated by the model. The reported discrepancies can be influenced by the input used to

185 force our limited-area simulations. This can be seen in the underestimation of the albedo of sea ice covered surface despite the prescribed surface albedo in the model that is in accordance with the observed sea ice albedo. This bias is, therefore, related to differences in sea ice fraction in the model and in the observation and indicates that the sea ice fraction in the ECMWF input data is too small.

### 3.2 Surface net irradiances and cloud radiative effect over sea ice and below clouds

This section explores the effect of clouds on the surface radiative budget in the ACLOUD observations and in our ICON simulations over sea ice. For that purpose, we, at first, look at net surface irradiance, which we further split into its solar and terrestrial components. To ensure comparability, despite obvious differences between the ICON simulations and ACLOUD observations described in subsection 3.1, we will restrict our comparison to situations where the model and the observations are within the same cluster of the two-dimensional histograms of surface albedo and surface net terrestrial irradiance at the

same time. To distinguish between those clusters, a situation is defined as cloudy if the net terrestrial irradiance at the surface is larger than -50 W m$^2$. Furthermore, a surface is classified sea ice covered, if the surface albedo is larger than 0.7 but less than 0.85, which is equivalent to the daily averaged maximum albedo value used in our adapted albedo parameterization. As we are interested in cloud (radiative) properties over sea ice covered surface, we will focus our evaluation on those situations. Furthermore, this cluster is appealing as most low-level flight sections were performed under these conditions.

In Figure 3, we compare observed and simulated net near surface irradiances using histograms. From Figure 3a, it becomes obvious that the model systematically overestimates net surface irradiances below clouds and over sea ice. This variable also

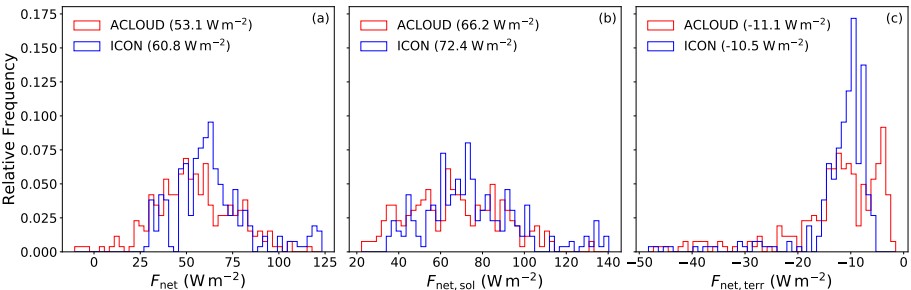

**Figure 3.** Relative frequency distributions of (blue) modeled and (red) observed surface net irradiation for sea-ice covered surfaces and cloudy conditions for (a) total radiation, (b) solar, and (c) terrestrial radiation. Values in the legend indicate the median of the respective variables.

shows a quite strong variability for both the model and the observations, which is related to varying sea ice albedo during the campaign. Additionally, the incoming solar radiation varied between research flights as they took place at different times of the day, which also introduces further variability. Looking at medians of the spectral components, we find that differences between simulated and observed net surface irradiances are mainly mediated by its solar component, while the median of net terrestrial surface irradiances are well simulated by ICON and also the shape of their histograms match better. Besides the above reported underestimated surface albedo for sea ice covered surface in ICON, also misrepresented cloud optical properties can contribute to the positive bias in net solar irradiances at the surface.

Furthermore, we investigate the surface cloud radiative effect (CRE) during ACLOUD, which is defined as the difference between net surface irradiance for cloudy and cloudless conditions. In the model, cloudy and cloudless irradiances can easily be derived by a double call to the radiation routines, one with clouds and one without clouds, leaving all variables not related to clouds constant. For observations, it is impossible to simultaneously observe both cloudy and cloudless conditions. Therefore, irradiances of cloudless conditions were obtained from dedicated radiative transfer simulations that used observations of atmospheric (i.e. temperature/humidity profiles) and surface properties (albedo). The one-dimensional, plane-parallel discrete ordinate radiative transfer solver DISORT (Stamnes et al., 1988) included in the libRadtran package (Emde et al., 2016) was applied for this purpose. The molecular absorption parameterizations from Kato et al. (1999) for the solar spectral range (0.28-4 $\mu$m) and from Gasteiger et al. (2014) for the terrestrial wavelength range (4-100 $\mu$m) were chosen. For calculating the observationally based CRE, the observed all-sky albedo was used, which also is used to create the prescribed functional dependency of the sea ice albedo that has been applied in the ICON model. Potential inconsistencies regarding the surface-albedo-cloud interaction and related issues discussed in Stapf et al. (2020) (they applied cloudless albedo estimates) are thus avoided. Unavoidable uncertainties in the comparison caused by the different applied radiative transfer schemes remain possible

The overwhelming majority of the observed and modeled total (solar plus terrestrial) surface CRE values are positive over sea ice, which indicates that clouds have a warming effect on the surface (Figure 4a). This is consistent with the relatively high surface albedo values at the onset of the melting period during ACLOUD (Jäkel et al., 2019; Wendisch et al., 2019), which

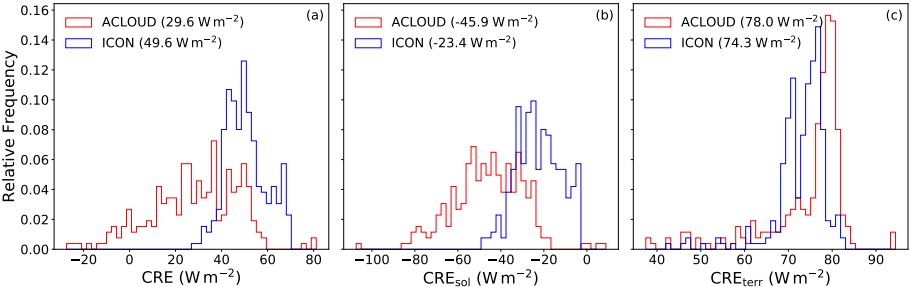

**Figure 4.** As Figure 3, but for the (a) total, (b) solar, and (c) terrestrial net cloud radiative effect at the surface.

decreases the cooling effect of clouds in the solar spectral range. Similar to the net surface irradiance, ICON overestimates the total surface CRE (Figure 4a), which is mainly caused by less cooling due to solar CRE (Figure 4b), while the modeled terrestrial CRE again matches the observed surface terrestrial CRE (Figure 4c). The way that the surface solar CRE is defined allows us to narrow down which effect is the main cause for the overestimated net solar surface irradiances. If clouds were perfectly simulated by the model, the negatively biased surface albedo would cause a too strongly negative surface solar CRE. As this is not the case for ICON, it is inferred that the main reason for the overestimated net solar surface irradiances is related to overestimated transmissivity of the cloud layer, which is defined as the ratio of downward transmitted solar irradiance at cloud base to downward incident solar irradiance at cloud top. Therefore, underestimated cooling effects in the solar spectral range are most likely related to incorrect simulations of microphysical or macrophysical properties of Arctic clouds in ICON. In the following section, we therefore compare those properties as they were simulated (ICON) and measured (ACLOUD) in more detail.

## 4  Comparison of macro- and microphysical cloud properties in ICON to ACLOUD observations

Transmissivity $T$ of a cloud layer is directly related to its optical thickness $\tau_c$:

$$T = \exp(-\tau_c),\tag{1}$$

where $\tau_c$ is defined as the volumetric cloud particle extinction coefficient $\beta_{ext}$, vertically integrated from cloud base $z_{base}$ to cloud top $z_{top}$:

$$\tau_c = \int_{z_{base}}^{z_{top}} \beta_{ext}(z)\,dz.\tag{2}$$

During ACLOUD and PASCAL, clouds were mostly in the liquid water phase with only little ice present, which allows to express the extinction coefficient as a function of liquid water content $q_c$ and cloud droplet number concentration $N_d$ (Grosvenor et al., 2018):

$$\beta_{ext} \sim N_d^{\frac{1}{3}} \cdot q_c^{\frac{2}{3}}.\tag{3}$$

Equation 3 and Equation 2 show that $\tau_c$ depends on geometrical depth ($z_{top} - z_{base}$), as well as on $q_c$ and $N_d$. In this study, we will denote the geometrical depth as a cloud macrophysical property and denote $q_c$ and $N_d$ as cloud microphysical properties. Nevertheless, we are aware that liquid water content, especially in a model that employs a saturation adjustment, cannot be considered to be solely a microphysical property as it strongly depends on the thermodynamical state of the atmosphere, thus making it a macrophysical variable that is adjusted by microphysical processes.

To identify potential sources explaining the model-measurement differences discussed in the previous section, we compare geometrical cloud thickness and microphysical properties of clouds in ICON to observations collected during ACLOUD/PASCAL. We decided to focus on the period from 2 June to 5 June 2017, when flights were possible on three out of four days. Here, only a brief summary of the meteorological conditions during that period is given. For a comprehensive overview of this period, we would refer the reader to Knudsen et al. (2018) and Wendisch et al. (2019). During this period, a southerly to easterly inflow of warm and moist air into the region where research flights took place was observed. Average near-surface temperatures and integrated water vapor at R/V Polarstern during that period were -3° C and $6\,\mathrm{kg\,m^{-2}}$, respectively. A relatively shallow, inversion-capped atmospheric boundary layer (Knudsen et al., 2018) with cloud top heights of less than 500 m in the vicinity of R/V Polarstern was observed. During those four days, the low-level cloud field was relatively homogeneous and mostly stratiform, with almost no high clouds being present in the domain where the research flights took place. Mostly liquid water and mixed-phase clouds were observed during this period (Wendisch et al., 2019). The relatively stable meteorological conditions during this period facilitated the statistical aggregation of the measurements all the research flights that took place during that period, which was not as straightforward for other parts of the campaign. Especially during mid June 2017, broken multi-layer clouds were present, which made a consistent comparison between the model and the observations harder to achieve. This can be seen in the limited amount of simultaneously cloudy and sea-ice-covered scenes in the period from 16 June to 18 June (see Table 1). Additionally, in-situ observation of cloud microphysical properties were performed on all flight days during that period. Another important point why this period was chosen is the fact R/V Polarstern was within the sea-ice-covered region and provided another source of observations that we can use for the comparison with our ICON simulations.

## 4.1 Geometrical cloud depth

We compare geometrical cloud depth as simulated by ICON to that observed during PASCAL. We choose PASCAL cloud radar and ceilometer observations instead of ACLOUD observations as they provide a continuous dataset in time, which facilitates the comparison of geometrical cloud depth. To better compare the simulations to ground based observations, we use ICON's meteogram output. It provides profiles of model variables at a certain location at every model timestep compared to the 30 minute output frequency when outputting the whole model domain. For each day simulated, we choose to output the profiles at Polarstern's 12 UTC location. While its position was rather constant from 3 June onward (Wendisch et al., 2019, their Figure 2), the ship was still in transit to the ice floe on 2 June. This might introduce some inconsistencies in the comparison to the spatially fixed ICON profiles. As the ship was already relatively far into the marginal sea ice zone, the cloud field should be homogeneous and representative for sea ice covered conditions.

For the model output, a layer within a profile is considered cloud covered if the total cloud condensate (liquid and ice) is larger

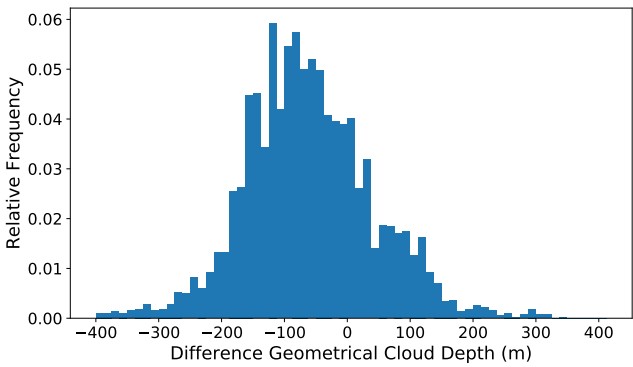

**Figure 5.** Difference in geometrical cloud depth between ICON and as observed from R/V Polarstern during the period from 2 June to 5 June.

than a threshold of $0.05\,\mathrm{g\,m^{-3}}$. We only assess clouds close to the surface, namely from the ground to 2 km altitude. In this altitude range, we define cloud base/top as the lowest/highest model level a cloud is being simulated within a profile. To derive the observed geometrical cloud depth, we use cloud base height as observed by the laser ceilometer on board R/V Polarstern while cloud top height was derived using the 35 GHz cloud radar (Griesche et al., 2019). Both modeled and observed cloud depths have been temporally interpolated to be on identical timesteps. We acknowledge that such a comparison of geometrical cloud thickness is not a definition aware comparison as it depends on instrument sensitivities and on the chosen threshold of total cloud condensate for diagnosing clouds in the model. Additionally, the rather simple approach is not able to correctly diagnose cloud depth for multi-layer clouds but as stated above, mostly single layer clouds were observed and simulated during the period of interest.

The difference in geometrical cloud depth simulated by ICON and as observed from R/V Polarstern during the period from 2 June to 5 June is shown in Figure 5. In general, the geometrical cloud depth is slightly negatively biased in our ICON simulations with a mean bias of 65 m and a standard deviation of 110 m. In offline radiative transfer simulations, we explored the effect of this bias in cloud geometrical thickness on the solar component of the surface CRE (see supplement). For that, we used profiles of liquid water that have been observed during the period from 2 June to 5 June and interpolated those profiles in the vertical. For all those profiles, a bias in 65 m in cloud vertical extent lead to change in solar CRE of approximately $5\,\mathrm{W\,m^{-2}}$, which is not sufficient to explain the reported model bias of more than $20\,\mathrm{W\,m^{-2}}$. Therefore, we will now focus on how cloud microphysical properties are represented in ICON compared to the observations and to which extent they contribute to the ascertained biases in cloud optical properties.

## 4.2 Cloud microphysical properties

To investigate how cloud microphysical properties contribute to the underestimated cloud optical thickness in ICON, we make use of the suite of in situ instruments that were part of the instrumentation of Polar 6 (Ehrlich et al., 2019). From 2 June to 5 June, research flights with Polar 6 were performed on three out of four days (no flight on 3 June). We focus on particle size

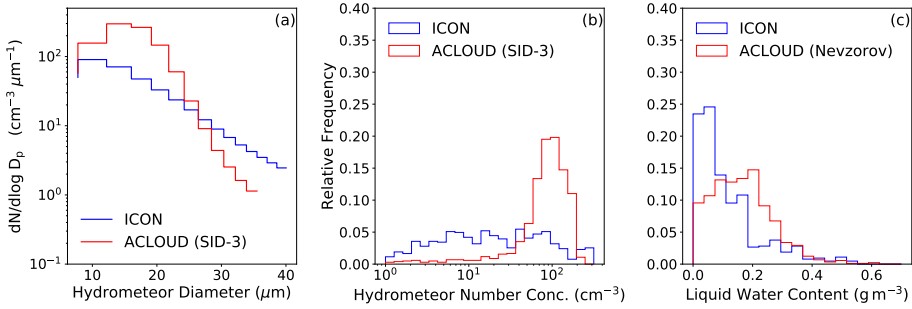

**Figure 6.** Time-space average particle number size distribution (a) and relative frequency of total particle number in the diameter range from 5 to 40 $\mu$m (b), as well as liquid water content (c). All data is averaged over the flights from 2 June to 5 June over sea ice covered region. Filtering for sea ice covered ACLOUD flight sections is done using simulated albedo from ICON.

distribution of hydrometeors and the respective moments, which have been observed by the Small Ice Detector Mark 3 (SID-3) covering a size range of cloud droplets/ice crystals, from 5 to 40 $\mu$m. As particle size distributions derived from SID-3 agree
well with those from other sensors (such as the Cloud Droplet Probe, CDP) for days when both probes were available (Ehrlich et al., 2019), we are confident that particle size distributions from the SID-3 are best suited for our comparison. In the following, we compare simulated and observed particle size distributions as well as the total particle number concentration ($N_{\mathrm{d}}$), mainly consisting of droplets in the size range presented in Figure 6. Furthermore, the liquid water content ($q_{\mathrm{c}}$) is shown. To be comparable to the particle size distribution from the SID-3, we integrate the size distribution of the two-moment microphysical
scheme implemented in ICON within the size bins of the SID-3 for cloud droplets and ice crystals, and add them. Due to relatively warm temperatures in the region of the research flights in early June 2017, only little ice was present in clouds during that period. While we derive the particle number concentration directly from particle size distribution by integrating over the size bins of the SID-3, we use measurements from the Nevzorov probe on Polar 6 to obtain information on $q_{\mathrm{c}}$.

Figure 6 shows particle number size distributions and the particle number concentration and liquid water content ($q_{\mathrm{c}}$) for
the period from 2 June to 5 June. Looking at the particle size distributions, we find that ICON underestimates the number concentration for hydrometeors smaller than 25 $\mu$m, while it overestimates the amount of cloud particles larger than that threshold in comparison to the measurements. As the number concentration of hydrometeors is mainly influenced by the number of small particles, the total amount of hydrometeors is also underestimated in the model. Averaged over all bins, $q_{\mathrm{c}}$ is underestimated by ICON relative to $q_{\mathrm{c}}$ derived by the Nevzorov probe, as the models overestimates the frequency of occurrence
for relatively small $q_{\mathrm{c}}$ values.

## 5 Discussion

### 5.1 Representation of cloud microphysical parameters in ICON

According to Equation 3, the underestimated hydrometeor number concentration and $q_c$ both can lead to lower cloud optical thickness in ICON. As not all microphysical schemes in ICON do provide number concentration of cloud droplets and ice crystals, the calculation of cloud optical properties is simplified in the radiation scheme. As an input for the radiation routines for liquid water clouds in ICON, a constant profile of $N_d$ is used, that decreases exponentially with altitude, and $q_c$ for the calculation optical properties of liquid clouds. For open water/sea ice, the assumed surface $N_d$ within the radiation scheme is $80 \, \mathrm{cm}^{-3}$, which is close to the observed cloud hydrometeor number concentrations (Figure 6). Nevertheless, this value is slightly lower than the observed mean of $85 \, \mathrm{cm}^{-3}$ for the three flight days from 2 June to 5 June. Assuming that the model is able to correctly simulate $q_c$, this underestimation would imply lower cloud optical thickness, which would further contribute to the overestimated amount of downward solar irradiance that reaches the surface. Calculation of optical properties of ice clouds is even further simplified as they depend solely on the ice water content. To evaluate the effect of cloud ice on radiative properties in the model, we performed a sensitivity analysis in which we turned off any radiative effect of cloud ice. This analysis revealed only a minor impact of cloud ice on radiation properties like surface CRE and net irradiance at the surface, which was both on the order of $1 \, \mathrm{W \, m}^{-2}$ compared to the basic set-up. This low impact is due to the already low cloud ice fraction in the model, which causes the radiative effect of cloud ice to be low. Due to the limitations of the observational dataset with little cloud ice being observed, it is hard to constrain the model from the observational side. Therefore, any estimation of the impact of cloud ice on the radiative balance has to be interpreted with some caution.

Additionally, $q_c$ in the model is underestimated compared to the observations, which also contributes to the bias in cloud optical thickness in ICON. We attribute the lower $q_c$ to an underestimated number concentration of relatively small cloud droplets (diameters $< 25 \, \mu\mathrm{m}$), which are commonly observed for this region and season (Mioche et al., 2017). The model also overestimates the number of hydrometeors with diameters larger than $25 \, \mu\mathrm{m}$. Thus, too few cloud droplets are generated and, therefore, condensational growth and coalescence of the available cloud droplets shifts the size distribution towards larger droplets. Looking at the phase state of precipitation reaching the surface in the region around R/V Polarstern ($81°-85°$ N and $5°-15°$ E), where most of the research flights from 2 June to 5 June took place, we find that rain rate at the surface ($8.57 \, \mathrm{g \, m}^{-2} \, \mathrm{h}^{-1}$) is almost an order of magnitude large than that of snow ($2.95 \, \mathrm{g \, m}^{-2} \, \mathrm{h}^{-1}$). As temperatures in the atmospheric boundary layer over sea ice were mostly below freezing during the three days analyzed, this rain must stem from "warm" rain processes, indicating an relatively active autoconversion process in our set-up. Therefore, autoconversion further contributes to the underestimated $q_c$ by ICON as it acts as a sink for cloud liquid water.

Interestingly, the here reported systematic underestimation of hydrometeors is different from the findings by Schemann and Ebell (2020). They conducted simulations for the Ny-Ålesund research station using the ICON model in the large-eddy set-up (ICON-LEM), and compare ground-based cloud radar observations with their ICON-LEM simulations applying a radar forward operator. Besides a different scheme for turbulent transport and activated parameterization of shallow convection in our set-up, as well as corresponding initial/boundary conditions from DWD's operational ICON forecast (instead of ECMWF

forecast), the basic set-up is similar to our simulations. Comparing radar reflectivities using contoured frequency by altitude diagrams in mid June 2017 (see Figure 6 in Schemann and Ebell, 2020), they found that for their 75 m domain, the model strongly overestimates the frequency of occurrence for low radar reflectivities/small hydrometeors. They argue that this finding can be related to the way CCN are activated into cloud droplets in the default Seifert-Beheng two-moment microphysical scheme. This was confirmed by ICON-LEM simulations in an Arctic domain by Mech et al. (2020) who implemented different CCN activation scheme (Phillips et al., 2008) within the Seifert-Beheng two-moment microphysics.

## 5.2 Revised activation of CCN in ICON

In the following, we will focus on the issue of the non-matching particle number size distribution compared to ACLOUD observations and how it affects total droplet number and $q_c$ of clouds in our simulations. As it has been pointed out by Schemann and Ebell (2020), this process might presently be misrepresented in the model. In its present implementation into ICON, the activation of CCN is parameterized as a function of grid-scale vertical velocity $\overline{w}$ and pressure $p$ as described in Hande et al. (2016):

$$\text{CCN}_{\text{act}} = A(p) \cdot \arctan\left[B(p) \cdot \log(\overline{w}) + C(p)\right] + D(p) , \tag{4}$$

where the parameters $A(p)$ to $D(p)$ contain information on the vertical profile of CCN and on the activation of CCN with respect to grid-scale vertical velocity $\overline{w}$. The profile presently used in the two-moment microphysical scheme is a temporally and spatially constant profile taken over Germany for a day in April 2013 as in Heinze et al. (2017). This CCN activation profile is not representative for the amount of CCN in the Arctic domain, as the CCN concentration in the Arctic is much lower. As stated in Schemann and Ebell (2020), the overestimated frequency of occurrence for low radar reflectivities/small hydrometeors in their simulations can be related to this unsuitable CCN profile.

Despite this unsuited CCN activation profile for an Arctic domain, we find an underestimated amount of hydrometeors in our simulations. Therefore, it is plausible that the relatively low hydrometeor number concentration is related to the coarser resolution in our ICON simulations. A realistic simulation of turbulence and cloud-scale vertical motion is crucial for Arctic mixed-phase clouds (Rauber and Tokay, 1991; Korolev and Field, 2008; Shupe et al., 2008). As the number of activated CCN is a function of grid-scale vertical velocity, it is likely that our simulations at 1.2 km resolution do not sufficiently resolve in-cloud vertical motion and turbulence (Tonttila et al., 2011). This is consistent with the fact that characteristic eddy sizes in Arctic mixed-phase clouds is less than 1 km (Pinto, 1998). Fan et al. (2011) suggested that only horizontal model resolutions of less than 100 m are able to resolve major dynamic features that contribute to vertical motion in Arctic mixed-phase clouds. Not being able to resolve those features consequently affects particle size distributions and its moments like number concentration as too few droplets are activated (Morrison and Pinto, 2005).

To account for subgrid-scale vertical motion, vertical velocity in the aerosol activation in larger scale models is often parameterized as a function of specific turbulent kinetic energy (Ghan et al., 1997; Lohmann et al., 1999), TKE, which is defined as:

$$\text{TKE} = \frac{1}{2} \cdot \overline{(u'^2 + v'^2 + w'^2)} , \tag{5}$$

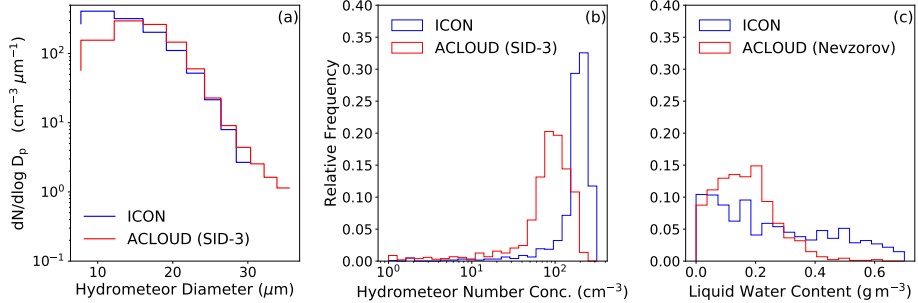

**Figure 7.** As Figure 6 but for the revised CCN activation. Due to different cloud fields in this simulation, the red lines (ACLOUD) are not identical with Figure 6 because of the sampling strategy employed as only datapoints in the observations and the simulation are being used if both are within a cloud simultaneously.

where the $u', v', w'$ are the subgrid-scale deviations from grid-scale velocitiy and the overbar denotes grid-box average. To explore the effects of including sub-grid scale vertical velocity in the Hande et al. (2016) CCN activation parametrization, we choose to follow a similar approach as proposed in Ghan et al. (1997), who assume the sub-grid vertical velocity in a grid box to follow a Gaussian distribution $P(w \,|\, \overline{w}, \sigma_w{}^2)$. The grid box averaged number of activated CCN can, therefore, be written as the integral over positive vertical velocities:

$$\overline{\mathrm{CCN_{act}}} = \int\limits_0^\infty P(w \,|\, \overline{w}, \sigma_w{}^2) \cdot \mathrm{CCN_{act}}(w) \, \mathrm{d}w \; . \tag{6}$$

To numerically solve the integral in Equation 6, a simple trapezoidal integration is employed using 50 equally spaced bins in a $\pm 3\, \sigma_w$ range around $\overline{w}$.

If it is assumed that sub-grid scale motion in low-level Arctic mixed-phase clouds is isotropic ($u'^2 = v'^2 = w'^2$), as proposed by Pinto (1998), the variance of vertical velocity can be expressed as function of TKE as follows (Morrison and Pinto, 2005):

$$\sigma_w{}^2 = w'^2 = \frac{2}{3} \cdot \mathrm{TKE} \; . \tag{7}$$

Using turbulence measurements on a tethered balloon during the PASCAL ice floe operations, Egerer et al. (2019) showed that isotropic turbulence is a valid assumption for a subset of days during PASCAL that have been analyzed in their study. We, nevertheless, are aware that isotropic sub-grid scale motion in Arctic clouds cannot be assumed for all conditions (Curry et al., 1988; Finger and Wendling, 1990).

The effects of this revised CCN activation for the period from 2 June to 5 June are shown in Figure 7. Compared to the original activation parameterization, the model shows a much closer agreement with the measurements, although an overestimation of hydrometeors with diameters less than $20\,\mu$m is simulated, while it underestimates the number of hydrometeors larger than $30\,\mu$m. As the number of small hydrometeors governs the total number of hydrometeors, their overestimation leads to an overestimated number of total hydrometeors in the whole diameter range between 5 and $40\,\mu$m. The particle size distribution now is in better agreement with the findings by Schemann and Ebell (2020), as we find an overestimation of smaller hydrometeors and

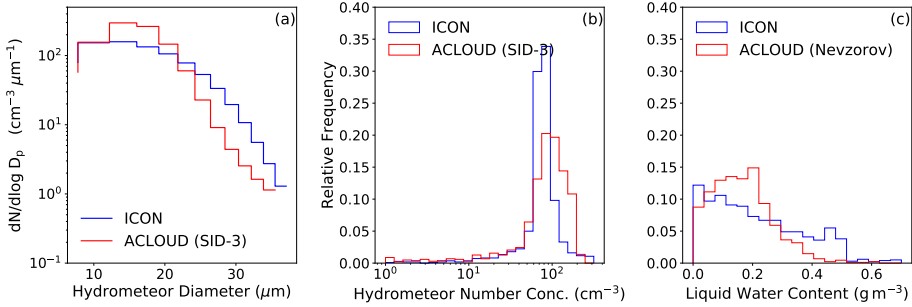

**Figure 8.** As Figure 7 but with scaled number of activated CCN by a factor of 0.4. Due to different cloud fields in this simulation, the red lines (ACLOUD) are not identical with Figure 6 and Figure 7 because of the sampling strategy employed as only datapoints in the observations and the simulation are being used if both are within a cloud simultaneously.

underestimated number concentration of larger hydrometeors compared to in situ observations. The shift of the particle size
distribution towards smaller hydrometeors can be related to the unsuited CCN profile within the activation parameterization.
As discussed above, autoconversion is the predominant sink for cloud water in the absence of precipitation formation via the
ice phase. The fact that the revised activation of CCN increases $N_d$ eventually leads to a reduction in the size of cloud droplets
(see Figure 7a). This reduces the collection efficiency of cloud droplets which leads to a less efficient autoconversion process,
which can be seen in the shift in the histogram of $q_c$ towards higher values in Figure 7c. Compared to the ACLOUD observa-
tions, small values of liquid water content less then $0.3\,\mathrm{g\,m^{-3}}$ are underestimated, while values larger than that threshold are
simulated more frequently in the revised CCN activation.

The presently used CCN activation profile was originally derived for spring conditions in Germany, where one would expect a
much higher load of CCN compared to the Arctic. To have a more realistic representation of CCN, a dedicated simulation with
a model that is able to represent the formation and transport of aerosols would be necessary. We opt against this approach and
instead scale the number of activated CCN from the default profile using a scaling factor of 0.4. A more elaborate description
why this scaling factor was used is given in Appendix A. The chosen scaling factor results in an underestimated number of
hydrometeors smaller than $22\,\mu$m as it is shown in Figure 8, while hydrometeors with larger diameters are overestimated by the
model. Looking at the hydrometeors number concentration, the chosen scaling factor shifts the simulated distribution towards
smaller hydrometeor concentrations that consequently results in a slight underestimation of hydrometeors compared to the
observations. This indicates that the chosen scaling factor is slightly too effective in reducing the number of activated CCN.
Compared to Figure 7, high values of liquid water content larger than $0.3\,\mathrm{g\,m^{-3}}$ occur less frequently when scaling the number
of activated CCN, but there is still a slight underestimation in the frequency of occurrence for $q_c$ values between $0.1\,\mathrm{g\,m^{-3}}$
and $0.3\,\mathrm{g\,m^{-3}}$. Even though scaled, the overall shape of the profile of activated CCN as a function of vertical velocity remains
unchanged. A different aerosol composition or just a different vertical profile of aerosols alter the shape of the profile, which
might also lead to biases in the number of activated CCN. This emphasizes the need for an CCN activation profile that is better
suited for an Arctic environment, which has also been proposed by Schemann and Ebell (2020).

The effect of the different CCN activation set-ups on the CRE for all flights from 2 June to 5 June is shown in Figure 9 (a)-(c). We would like to point out that the cloud fields between the respective CCN activation set-ups vary. For that reason, the number of available datapoints for which the threshold for sea ice coverage and cloudy conditions are fulfilled at the same time, differ
between the runs due to the filtering that is employed. Similar to the histograms in Figure 4, which cover all flights used in this comparison, the warming effect of clouds at the surface is overestimated when looking at the period from 2 June to 5 June. For the revised CCN activation, the increase in $q_c$ reflects on the surface CRE, which now has a small negative bias compared to the ACLOUD observations. Because of the aforementioned constant profile of cloud droplet number concentrations in the calculation of the effective radius within the radiation scheme, this negative bias would be more strongly expressed if the actual
cloud droplet number concentration from the microphysical scheme would be used (see subsection 5.3). When scaling the activated number of CCN by a factor of 0.4 using the revised CCN activation, the CRE is still overestimated by ICON compared to observations even though the positive bias in the median could be reduced by approximately $5\,\mathrm{W\,m^{-1}}$. As downscaling the number of activated CCN by a factor of 0.4 was already slightly too effective in reducing the hydrometeor number, a larger scaling factor might be able to further decrease the CRE in the model.

From the previously conducted sensitivity study employing a more effective CCN activation, it is not clear whether the above reported biases in cloud microphysical properties is a source (inefficient CCN activation) or a sink issue (too effective autoconversion). To this end, we conducted a further sensitivity study with unchanged CCN profile and in which autoconversion was turned off entirely (see supplement). While the effect on $q_c$ is comparable to the revised, but not yet scaled CCN activation (see Figure 7), the cloud droplet number concentration is still underestimated. Furthermore, the shape of the size distribution does
not match the shape of the observed one. Since the CCN profile used in the activation of CCN into cloud droplets within the cloud microphysical scheme is not suited for an Arctic domain as it overestimates the availability of CCN, the underestimated amount of cloud droplets in the simulations with autoconversion turned off is indicative for a source rather then a sink problem of cloud droplets in our simulations.

## 5.3 Coupling of hydrometeor number concentration to radiation

As already discussed above, there is an inconsistency between the hydrometeor number concentration derived in the two-moment microphysics and used in the radiation routines. In the following, we therefore explore the effect of making the hydrometeor concentrations consistent between the two parametrizations. As input for the calculation of optical properties, ICON uses cloud droplet/ice crystal effective radius, which is defined as the ratio of the third to the second moment of the size distribution. Previously, effective radii were computed as a function solely of specific masses.
To ensure consistency with the size distributions in the Seifert-Beheng two-moment scheme, we calculate the effective radii from the used gamma distribution (see Appendix B for the derivation). This new implementation has already been used in Costa-Surós et al. (2020). In Figure 9 (d)-(f), the biggest difference to the uncoupled hydrometeor number concentrations (Figure 9 (a)-(c)) can be seen in the histograms for the revised CCN activation (Figure 9 (e)). In this set-up, the CRE is underestimated compared to observations due to higher hydrometeor concentration, which is now also considered in the radiation
parameterization. For the revised and scaled CCN activation, only little differences are simulated between coupled and un-

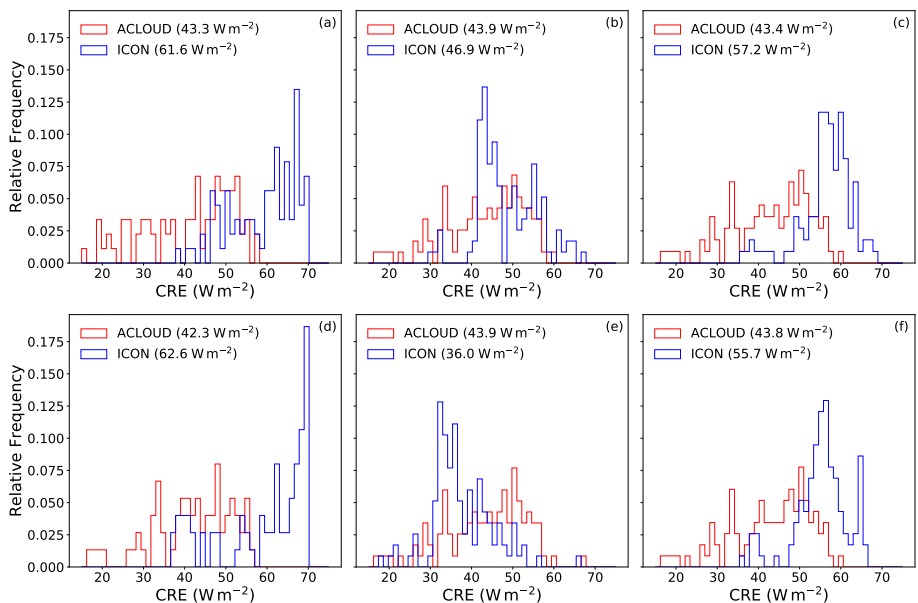

**Figure 9.** As Fig. 4a, but for the flights from 2 June to 5 June only, for the default set-up (a), for the revised CCN activation (b) and for the revised CCN activation with scaled number of activated CCN by a factor of 0.4 (c). The bottom row (d-f) as the top row but with hydrometeor number concentration coupled to radiation. Due to different cloud fields in the respective simulations, the histograms for the ACLOUD observations are not identical as only datapoints in the observations and the simulation are being used if both are within a cloud simultaneously.

coupled hydrometeor concentration. As stated above, the fixed cloud droplet number concentration in the default radiation routines is already relatively close to the hydrometeor concentration observed for the flights from 2 June to 5 June. Nevertheless, compared to the observations, the median value of the CRE in ICON in Figure 9 (f) is closest to the observed values, even though they are still slightly overestimated. Altogether, the revised CCN activation with a scaled CCN activation and

coupled hydrometeor now results in a positive bias of only approximately $6\,\mathrm{W\,m^{-2}}$. The effect on surface CRE of the coupling of hydrometeor number concentration to radiation for this period is relatively low ($1\,\mathrm{W\,m^{-2}}$, see Figure 9 (c) and (f)), as the assumed number concentration in the default set up and the number concentrations from two-moment microphysical scheme in the revised and scaled CCN activation are in a similar range. As can be seen from Figure 9 (b) and (e), if the $N_d$ profile in the microphysics deviates from the profile in the radiation, there can be quiet substantial differences due to a more realistic

representation of the Twomey effect (Twomey, 1977), which can be important for relatively clean/polluted situations. As it can be seen in Figure 4, the differences in the CRE for the respective sensitivity experiments are again primarily mediated by its solar component, whereas the terrestrial components are in good agreement with the observationally derived terrestrial CRE components (see supplement).

## 6 Conclusions

In this study, we use observational data from the ACLOUD and PASCAL campaigns (Wendisch et al., 2019) to compare it to limited-area simulations with the ICON atmospheric model at kilometer-scale resolution. While the model compares well to the observations in its ability to simulate the four cloud-surface radiation regimes in the Arctic, it severely underestimates cloud radiative effects in the solar spectral range. This is despite a slight underestimation of the geometrical cloud thickness and attributable to too small droplet number concentrations and too little liquid water content simulated by the model. We showed that it is crucial to correctly represent in-cloud turbulence in Arctic clouds, which is essential to correctly simulate hydrometeor number concentration and liquid water content. The findings of this study are mainly representative in the case of turbulence driven, stratiform and optically thin single-layer clouds that contain liquid water but are, to some extent, also valid for multi-layer clouds, which was confirmed by an analysis of days in mid June 2017, where such conditions prevailed. Furthermore, similar improvements were obtained at lower horizontal and vertical resolution (2.4 km and 50 vertical levels) when including sub-grid vertical motion in the activation of CCN into clouds droplets, which makes us confident that such an approach can also be beneficial for simulations with coarser spatial resolution.

As reported by Stevens et al. (2020), the representation of clouds in atmospheric models benefits from higher resolved simulation. Nevertheless, long time, global simulations at hectometer scale will not be feasible in the foreseeable future (Schneider et al., 2017), whereas climate projections at kilometer-scale can be achievable (Stevens et al., 2019). It is, therefore, important to especially improve models on such scales to enable them to make realistic simulations. As shown in this study, aircraft observations are a valuable source of information and can be used for evaluating and improving the representation of physical processes for models at kilometer-scale. The results presented in our study might also be beneficial to the representation of clouds in ICON in other regions, where clouds are also turbulence-driven.

*Data availability.* The ICON model output data used in this study is stored at the German Climate Computing Center (DKRZ) and is available upon request from the corresponding author. The observational data from the ACLOUD/PASCAL campaigns archived on PANGAEA repository and can be accessed from the following DOIs: broadband (solar and terrestrial) irradiances (https://doi.org/10.1594/PANGAEA.902603, Stapf et al., 2019), Small Ice Detector Mark 3 (SID-3) (https://doi.org/10.1594/PANGAEA.900261, Schnaiter and Järvinen, 2019), Nevzorov probe (https://doi.org/10.1594/PANGAEA.906658, Chechin, 2019) and cloud radar 35 GHz cloud radar onboard of R/V Polarstern (https://doi.org/10.1594/PANGAEA.899895, Griesche et al., 2019).

## Appendix A: Scaling of the default CCN profile

In this study, we decided to scale to default CCN profile in ICON to match values representative for the Arctic. The scaling factor is derived from aerosol mass mixing ratios from the re-analysis of atmospheric composition of the Copernicus Atmospheric Monitoring Service (CAMS; Inness et al., 2019), which assimilated MODerate Resolution Imaging Spectroradiometer (MODIS) aerosol retrievals (Levy et al., 2013) into the ECMWF model (Benedetti et al., 2009). We computed the number of

**Table A1.** Scaling factor that minimizes the mean squared error of the scaled default activation profile in ICON and the activation profile derived from CAMS for several vertical velocities in an altitude band from the surface to 700 hPa.

| $w$ (m s$^{-1}$) | 0.01 | 0.03 | 0.08 | 0.22 | 0.60 |
|---|---|---|---|---|---|
| Scaling factor | 0.5 | 0.4 | 0.4 | 0.4 | 0.3 |

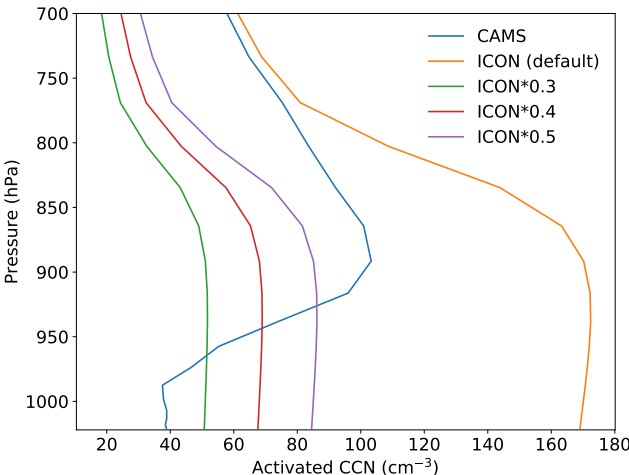

**Figure A1.** Profile of activated CCN at 0.08 m s$^{-1}$ from CAMS and from the default profile in ICON. Additionally, a subset of scaled ICON profiles is shown.

activated CCN for various vertical velocities and also supersaturation for a sea ice covered domain north of Svalbard during the period from 2 June to 5 June following the approach of Block (2018). Close to the surface, the number of activated CCN at a supersaturation of 0.5 % in this dataset is approximately 45 cm$^{-3}$. This value is on the lower end of the observed number concentrations of activated CCN during PASCAL, which were in a range of 40 to 80 cm$^{-3}$ during this period (Wendisch et al., 2019, their Figure 10).

To decide which scaling factor to use, we looked for a scaling factor (in steps of 0.05) that minimizes the mean squared error of the scaled profile and the profile derived from CAMS for several vertical velocities in an altitude band from the surface to 700 hPa. From Table A1, we find that a scaling factor of 0.4 is a good compromise for relatively low vertical velocities in Arctic clouds. Even though scaled to best match the CAMS profile, the overall shape of the profile of activated CCN in ICON remains unchanged. Figure A1 shows that the default profile strongly overestimates the number of activated CCN close to

the surface while nicely matches the CAMS profile for altitudes higher than 800 hPa. As almost all clouds from 2 June to 5 June were below that altitude, it is more important to correctly represent the number of activated aerosol close to the surface. The number of activated CCN is almost constant up to 850 hPa, whereas the number of activated CCN in the CAMS profile increases with altitude. Even though we cannot match the shape of the activation profile, a scaling factor of 0.4 should represent an approximate average up to 850 hPa.

## Appendix B:  Derivation of effective radius from gamma distribution

To describe the particle size distributions of all hydrometeor categories in the Seifert-Beheng two-moment microphysical scheme (Seifert and Beheng, 2006), a modified gamma distribution is used:

$$f(x) = A x^{\nu} \exp\left(-\lambda x^{\mu}\right),$$ (B1)

where $x$ is the particle mass and $\nu$ and $\mu$ are the parameters of the distribution for the respective hydrometeor category. $A$ and $\lambda$ can be expressed by the number/mass densities and the parameters $\nu$ and $\mu$ (Eq. 80, Seifert and Beheng, 2006). Following Petty and Huang (2011), the $k$-th moment $M_k$ of such a modified gamma distribution can be expressed as follows:

$$M_k = \frac{A}{\mu} \frac{\Gamma\left(\frac{\nu+k+1}{\mu}\right)}{\lambda^{(\nu+k+1)/\mu}}.$$ (B2)

The ration between 3th and 2th moment can, therefore, be written as:

$$\frac{M_3}{M_2} = \frac{\Gamma\left(\frac{\nu+4}{\mu}\right)}{\Gamma\left(\frac{\nu+3}{\mu}\right)} \lambda^{\frac{-1}{\mu}}.$$ (B3)

To obtain the effective radius, Equation B1 has to be first converted into a function of radius. According to Eq. 54 in Petty and Huang (2011) the particle size distribution as a function of radius $f(r)$ can be written as:

$$A_r r^{\nu_r} \exp\left(-\lambda_r r^{\mu_r}\right) = A x(r)^{\nu} \exp\left[-\lambda r^{\mu}\right] \frac{\mathrm{d}x}{\mathrm{d}r},$$ (B4)

The particle mass as a function of radius $x(r)$ in the Seifert-Beheng two-moment microphysical scheme is defined as follows:

$$x(r) = \left(\frac{2r}{a}\right)^{\frac{1}{b}},$$ (B5)

which differs from the functional relationship given in Table 1 in Petty and Huang (2011), as the values for $a$ and $b$ are defined differently (see Table 1 in Seifert and Beheng, 2006). Therefore:

$$\frac{\mathrm{d}x}{\mathrm{d}r} = \left(\frac{2}{a}\right)^{\frac{1}{b}} \frac{1}{b} r^{\left(\frac{1}{b}-1\right)}.$$ (B6)

Inserting Equation B5 and Equation B6 into Equation B4 and comparing the respective parameters for radius and mass in Equation B1, we find the following conversion relationships for the parameters in the particle size distribution:

$$A_r = \frac{A}{b}\left(\frac{2}{a}\right)^{\frac{\nu+1}{b}}, \quad \nu_r = \frac{\nu+1-b}{b}, \quad \lambda_r = \lambda\left(\frac{2}{a}\right)^{\frac{\mu}{b}}, \quad \mu_r = \frac{\mu}{b}.$$ (B7)

By inserting those parameters into Equation B3 and applying the functional dependencies for $A$ and $\lambda$ from Eq. 80 in Seifert and Beheng (2006), the effective radius $r_{\text{eff}}$ can be written as follows:

$$r_{\text{eff}} = \left[\frac{\Gamma\left(\frac{\nu+1}{\mu}\right)}{\Gamma\left(\frac{\nu+2}{\mu}\right)}\right]^b \left(\frac{q}{N}\right)^b \frac{a}{2} \frac{\Gamma\left(\frac{\nu+1+3b}{\mu}\right)}{\Gamma\left(\frac{\nu+1+2b}{\mu}\right)},$$ (B8)

where $q$ and $N$ are the mass and number density for the respective hydrometeor category.

*Author contributions.* JK, JS, MW and JQ this conceived this study. DK helped setting up the input data for the ICON runs and gave valuable expertise on how to run the model in a limited area set-up. JK and JS prepared and analyzed the model and observational data, respectively. All of the authors assisted with the interpretation of the results. JK prepared the manuscript with contributions from all co-authors.

*Competing interests.* The authors declare that they have no conflict of interest.

*Disclaimer.* TEXT

*Acknowledgements.* We gratefully acknowledge the funding by the Deutsche Forschungsgemeinschaft (DFG, German Research Foundation) - 268020496 - TRR 172, within the Transregional Collaborative Research Center "ArctiC Amplification: Climate Relevant Atmospheric and SurfaCe Processes, and Feedback Mechanisms (AC)[3]. The ICON model is jointly developed by German Weather Service (DWD) and the Max Planck Institute for Meteorology, Hamburg, and we thank the colleagues for making the model available to the research community. We furthermore thank the colleagues that participated in the ACLOUD/PASCAL campaigns for providing the datasets used in this study. 560 Simulations were conducted at the German Climate Computing Center (Deutsches Klimarechenzentrum, DKRZ). We thank Axel Seifert and Kerstin Ebell for giving valuable comments on this manuscript. We furthermore thank the two anonymous reviewers for their constructive comments.

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
