# Peer review of "Employing airborne radiation and cloud microphysics observations to improve cloud representation in ICON at kilometer-scale resolution in the Arctic"

_Atmospheric Chemistry and Physics, 2020_

## Referee Comment (RC1) · Anonymous Referee #1 · 30 Jul 2020

**Review of "Employing airborne radiation and cloud microphysics observations
to improve cloud representation in ICON at kilometer-scale resolution in the Arctic"**

by Kretzschmar et al.

submitted to Atmos. Chem. Phys. Disc.

Summary

This paper presents an analysis of surface radiative biases in kilometre-scale ICON simulations as compared to field observations obtained during the ACLOUD campaign which was conducted May-June 2017 around the region of Svalbard. Measurements were obtained over partial-sea ice and full sea-ice covered surfaces. Biases in surface solar and terrestrial irradiance within standard ICON configurations are attributed in this study to a misrepresentation of the surface albedo in short-time-scale simulations and an overestimation in cloud transmissivity for low-level clouds above sea ice. The latter is further attributed to an underestimation in cloud-droplet number concentration of small to medium-sized cloud droplets (10-25 μm), which also leads to an underestimation of cloud water content. In sensitivity experiments the authors demonstrate that a more accurate description of the activation process in kilometre-scale ICON simulations, and an adjustment of the background CCN profiles to Arctic conditions, decreases cloud transmissivity and thus improves the simulated cloud-radiative effect (CRE). The authors also show that an active coupling of the two-moment microphysics scheme to the ICON RRTM radiation scheme does not yield a considerable improvement of the simulated net CRE in this case.

Recommendation

This study presents a comprehensive evaluation of the CRE of low-level Arctic clouds in ICON simulations above sea-ice covered surfaces. Low-level clouds and in particular mixed-phase clouds impact the surface radiative balance substantially in this region and are often miss-represented in climate models. This analysis is thus addressing one of the key concerns within the community and will be of interest to a wide readership.
The paper is really well written and very logically structured. Their results are presented clearly and concisely and I agree with their scientific conclusions. Occasionally their arguments could be strengthened, which I point out in my comments below. Overall, I think this is an excellent paper that deserves publication once these minor revisions are addressed.

General comments

1. I understand that the case descriptions etc. are given in other papers. Yet from this paper it is not clear for which conditions you have tested the TKE-based activation approach and its impact on net CRE and for which conditions you associate the largest biases. While detailed case descriptions are not necessary, context should be given for the reader in terms of the conditions of June2-June5th (for which the bias attribution and sensitivity analysis is done). In particular information with respect to temperature regime, integrated water vapour content, optical depth regime, precipitation characteristics and stability would be useful. Also are these predominantly stratiform or broken cloud-decks? I would also suggest to contextualise your findings in the discussion section in terms of how far you would be comfortable to extrapolate your findings beyond the optically thin (I assume), single-layer cloud regime that you explored here in greater detail.

2.  You argue the utility and necessity to evaluate and improve kilometre-scale simulations. In this paper you provide a pathway to improve the simulated net CRE for "kilometre-scale" ICON simulations for Arctic low-level clouds, which may even yield to improvements to similar cloud regimes simulated in other regions of the globe. In order to use this approach more widely, it would be helpful to be aware of its potential limitations within the range of "kilometre-scale" grids. Here, you show results of a particular configuration of horizontal (1.2km resolution) and vertical resolution. Terms like "kilometre-scale", "convection-permitting", etc. are often used in the community for a range of resolutions ranging from, say, 1-5 km, which apply all kinds of vertical grid refinement within the boundary layer. How valid do you expect your conclusions to remain across the range of spatial resolutions that fall under the category "kilometre-scale"/"convection-permitting"? Would you expect your TKE fix to droplet activation to work equally well at a (say) 5km grid spacing, or when only half the vertical grid spacing is applied?

3.  In section 3 during your evaluation of surface radiative quantities you argue that you can compensate for the temporal irregularity of your model output (every 3h) by the increased spatial coverage and thus increased sampling of spatial variability. This essentially assumes that spatial and temporal variability are equivalent. This is assumption is commonly made during simulation-observation comparisons. Can you demonstrate this to be valid though for radiative quantities subject to a diurnal cycle?

4.  Your analysis of biases regarding net CRE is focused on the period of 2-5th of June. In L234 you state that you select this period because you largely are dealing with single-layered low-level clouds and have a high density of flights. I am assuming that the bias in CRE (Fig. 4) is also largest during this time period and for this particular cloud regime as well? Given the significance of the analysis that follows for the overall manuscript, I would include a couple more sentences on this selection for clarity.

5.  I agree with your general sentiment conveyed in the introduction and conclusion sections of this manuscript that high-resolution LES simulations are quite limited in their spatial and temporal coverage and that coarser-resolution simulations allow longer-term evaluations over larger domains. Yet I wonder, if you are not subject to the same limitations in this particular application, since you are restricting this evaluation to the location of 15 linear flight tracks within a particular region (although admittedly you can afford to simulate more flight hours), and most of your analysis is focused on the period June 2nd-5th. The argumentation of the benefits and limitations of kilometre-scale versus LES simulations does not seem an essential part of your analysis. Thus I would consider to reduce the emphasis on this point, as this is not something you actually show.

6.  Fig. 6 very clearly shows the bias in simulated cloud properties that are consistent with an overestimation in cloud transmissivity. From the observations you are under constrained and cannot (I presume) say with certainty whether this is a source or sink issue. In your analysis you show that the bias can be fixed by increasing the source in Nd. Can you provide an equally strong argument, that you could not obtain the same improvement, by adjusting the sink? I think this could be done in the context of a discussion of cloud-base or surface precipitation rates, or a couple of additional numerical experiments where you explicitly show that adjustments to the autoconversion rate by: either turning it off altogether – essentially shutting off warm rain – or reducing its efficiency, does not yield the same kind of improvement.

Specific comments

1. L44: This seems like a somewhat random selection of LES studies in the Arctic and by no means complete. I suggest to either include a comprehensive list of references, or to make it clear that this list of studies is merely exemplary.

2. L48ff: In addition to the representation of in-cloud turbulence and cloud-top inhomogeneity, LES setup also allows the study and evaluation of microphysical processes (e.g. Ovchinikov et al (2014), Solomon et al (2015), Fridlind et al (2017)) and aerosol-cloud interactions (e.g. Possner et al (2017), Solomon et al (2018), Eirund et al (2019)) at scales where the dynamics and thermodynamics are largely resolved. Since you identify the representation of CCN and the activation process itself as one of your primary sources of bias regarding net CRE. It seems fair to mention this here.

   Refs:
   1. Ovchinikov et al (2014): doi:10.1002/2013MS000282 (JAMES)
   2. Fridlind et al (2017): doi:10.1029/2007JD008646 (JGR)
   3. Solomon et al (2015): doi:10.5194/acp-15-10631-2015 (ACP)
   4. Possner et al (2017): doi:10.1002/2016GL071358 (GRL)
   5. Solomon et al (2018): https://doi.org/10.5194/acp-2018-714 (ACP)
   6. Eirund et al (2019): https://doi.org/10.5194/acp-19-9847-2019 (ACP)

3. L102: What is your reasoning for using the all or nothing cloud-cover scheme? Did it impact your results?

4. L132: "The daily averaged observed albedo is parameterized as a function of day of the year". I did not follow this. Did you not simply prescribe the daily mean albedo value from the full sea-ice covered surface observations. So how is it a "function" of the day of year?

5. L177/178: Why did you not fix the sea ice fraction in a similar fashion as the sea ice albedo in your simulation setup to exclude the impact of biases from essentially prescribed surface properties?

6. L217: I agree with your conclusion that the underestimated cooling in the solar spectral range is likely due to an incorrect simulation of cloud transmissivity, rather than remaining biases in surface albedo. As this a central aspect to your overall argument, I was wondering if you could not show this explicitly. Do your conclusions remain the same if you restrict the phase space your analysis of the observations to surface albedo values < 0.8 such as to match the simulations?

7. L231: I personally would argue that cloud water content is to first order a thermodynamic variable and thus also a macrophysical variable that is adjusted by microphysical processes (i.e. the efficiency of autoconversion/accretion in warm-phase clouds anyway). Especially in a model with saturation adjustment I have a hard time referring to qc as purely microphysical, but can be convinced.

8. "it shows a slight underestimation". Can you quantify this? What is the average/median cloud depth?

9. L260: I would suggest to be more specific/quantitative here, as this is a key argument in your assertion that the bias stems predominantly from biases in cloud water content and droplet concentration. For the typical cloud optical depth seen in your simulations or during the observation record, how large would a geometrical cloud depth bias have to be to affect transmissivity substantially? How does that quantity relate to your biases assessed?
In that context, I am not sure Fig. 5 is best suited. I wonder if a PDF-based comparison is not more informative. Cloud transmissivity is strongly non-linearly related to geometrical cloud depth. Thus biases in the distribution of geometrical depth (although means may agree), could induce substantial biases in mean cloud transmissivity. To follow your line of assertion, the argument that cloud depth biases are unlikely to contribute signficantly should be strengthened quantitatively.

10. L270: "droplets plus ice crystals": essentially droplet concentration at >1cm-3. I would not expect to see any impact of ice in the shown range. It may be worth to state explicitly.

11. L300: Can you provide a quantitative estimate of rain rate? Do you have any constraint here from the observation?

12. Fig 2: In the range of surface albedos between 0.6-0.8 where the number of occurrence is highest, the simulations show a much considerably narrower range of Fnet,sol than the observations. Do you have any idea whether this is indicative of a model bias, or simply a sampling issue between the simulation and observation datasets?

13. Fig5: I personally find it hard to draw quantitative conclusions from this plot going beyond the overall range of values in observations and simulations. You can sort of see that geometrical cloud depth is likely underestimated, but its hard to tell due to the many overlapping points where the real density of points is. As suggested previously, I wonder if a PDF comparison would not be more informative

14. Fig. 6-8: Panel a): I am fairly sure the ICON and ACLOUD lines are swapped? Otherwise there would be a mismatch between your figure and the discussion and the results of sections 4.2ff. Panel b): The ACLOUD in Fig6 is slightly different to 7/8. Why?

15. Fig. 9: Ultimately the net cloud-radiative effect is of interest, but your argument primarily relates to CRE_sol. What is the impact of CRE_sol alone?

16. Table1: I personally would find a a total number of included flight hours as part of the caption helpful.

Edits

1. L165: suggested rephrase "to the previous comparison" to "to the previously used model setup".
2. L248: "the the"
3. L291: suggest rephrase "than observed the mean of" to "than the observed mean of"
4. L423: Typo? Underestimation of geometrical cloud depth, right?
5. L424: "represent" instead of "simulate" (since you do not really simulate it)

---

## Referee Comment (RC2) · Anonymous Referee #2 · 31 Jul 2020

In this paper, the authors compare simulations using the ICON model to observations from the ACLOUD and PASCAL campaigns. They find that the ICON simulations predict a more strongly positive cloud radiative effect (CRE) than that derived from the ACLOUD observations. They then determine that an important contribution to this difference is the small number of cloud condensation nuclei (CCN) activated in the ICON model, which subsequently results in low cloud liquid water contents. They improve the model results by accounting for the effects of subgrid-scale turbulence on cloud droplet activation and by scaling their assumed CCN profile. I feel that the study merits

publication, provided that the following comments are addressed:

General comments:

1. The authors briefly mention cloud ice in a few places in the paper, but they largely restrict their analysis to liquid cloud water. Some definitive or quantified statements about the contributions of ice clouds to the radiation balance or hydrometeor concentrations, both in ICON and in the observations, would be welcome. Could differences in the amount of frozen cloud make a significant contribution to differences in the surface radiation balance or the cloud radiative effect between the model and the observations?

2. Sect. 3.2, p9: The authors mention here that the CRE is calculated from the observations through the methods of Stapf et al. (2019a). Given that there are potential inconsistencies in the calculated CRE between the model and the observations, just a little more detail on the radiative transfer simulations of Stapf et al. (2019a) seems prudent here.

The authors mention that "While the prescribed functional dependence of the sea ice albedo has been derived for cloudless and cloudy conditions, the surface albedo that is used to derive the CRE from the observations is for cloudy-sky only. This can lead to inconsistencies between the modeled and observed CRE (Stapf et al., 2019a)." However, If I understand correctly, the radiative transfer simulations of Stapf et al. (2019a) account for cloud-surface-albedo interactions. Given that the surface albedo is prescribed in the ICON simulations, these cloud-surface-albedo interactions will not be accounted for in the ICON simulations. Therefore, wouldn't it be a more consistent comparison if the cloud-surface-albedo interactions were also neglected in CRE calculations based on the observed data? Can the authors comment on this?

Specific comments and technical corrections:

p2, line 38: optical -> optically

p2, lines 44-47: Please improve the clarity of this sentence.

p3, line 74: sea ice covered -> sea-ice-covered

p3, line 83: unmatched parenthesis: "given (for"

p3, line 84: refer -> refer the reader to

p5, line 108: "general feature of ICON." Perhaps the authors mean "generally representative of ICON"?

p5, line 120: "caused by the way how our simulations" Please either choose "the way that" or "how".

p8, line 176 "sea ice covered surface". This should be either "the sea-ice-covered surface" or "sea-ice-covered surfaces".

p8, line 189: "Figure 3 a" -> "Figure 3a"

p9, line 200: Please insert a comma after "without clouds"

p9, line 202: "measurements of atmospheric/surface observations" Perhaps the authors mean "atmospheric or surface measurements" or "atmospheric or surface observations"?

p9, line 211: Please either choose "The way that" or "How".

p9, line 212: "allows to narrow down, which effect" -> either "allows us to narrow down which effect" or "allows one to narrow down which effect".

p9, line 212: "If clouds would be" -> "If clouds were"

p9, line 215: "fraction" -> "ratio"

p10, line 226: "which allows to" -> "which allows us to"

p11, line 260: "extend" -> "extent"

p13, line 301: large -> larger

p13, line 302: stems -> stem

p14, lines 327-328:The overestimation of small hydrometeors mentioned here seems to be in contradiction to the statements of p12, lines 278-280.

p16, lines 393-394: Since the last simulation discussed was not the default set-up but instead was the one using the revised CCN activation scheme, most readers would assume that the authors are comparing the simulation with the CCN scaled by 0.4 to the revised CCN activation simulation. The authors need to make it clear that they are comparing this simulation to the default set-up.

p17, lines 411 and 414: Do the authors mean Figure 9f instead of 9e?

p20, eq. B3: If I divide eq. B2 with k=3 by eq. B2 with k=2, I find the trailing factor to be $A^{-1/\mu}$, not $\lambda^{-1/\mu}$. Is the error in eq. B2 or eq. B3?

Figure 1 caption: "inner domain has a" -> "inner domain (red) has a"

Figure 5 and p11, lines 258-261: There is significant overlap in the points on this plot, which makes it difficult to tell, for example, how large a fraction of the data have observed cloud depth < 0.4 and modelled cloud depth < 0.2. This also means that it is difficult to judge the degree of underestimation of the cloud depths. I don't have a perfect solution for this issue, but the authors may wish to consider making the data points partially transparent, or substitution of the scatter plot with a histogram (with different subplots for the different observation days, if the authors wish). I am open to other solutions, or arguments from the authors in favour of the current plot. In any case, the median values of the modelled and observed cloud depths should be provided to help the reader quantify the degree of underprediction. The means and standard deviations may also be helpful.

Figure 6, Figure 7, and Figure 8: The red lines for panels b and c are very similar in the three figures, but not quite identical. Note for instance that the peak in frequency of hydrometeor number concentration is > 100 in Figure 6 and < 100 in Figure 7 and

Figure 8. Rather than state in the captions that the lines are identical, the authors instead should very briefly remind the reader why the lines differ slightly. Also, it seems that the red and blue lines are reversed in panel a in all three figures.

Figure 9: It would be prudent to remind the reader in the caption that the red lines differ slightly due to the sampling that is applied.

---

## Author Comment (AC1) · 18 Sep 2020

**Response to referee comment #1**

*This study presents a comprehensive evaluation of the CRE of low-level Arctic clouds in ICON simulations above sea-ice covered surfaces. Low-level clouds and in particular mixed-phase clouds impact the surface radiative balance substantially in this region and are often miss-represented in climate models. This analysis is thus addressing one of the key concerns within the community and will be of interest to a wide readership. The paper is really well written and very logically structured. Their results are presented clearly and concisely and I agree with their scientific conclusions. Occasionally their arguments could be strengthened, which I point out in my comments below. Overall, I think this is an excellent paper that deserves publication once these minor revisions are addressed.*

We thank the reviewer for the constructive comments that helped to improve the manuscript.

**General comments**

*1. I understand that the case descriptions etc. are given in other papers. Yet from this paper it isnot clear for which conditions you have tested the TKE-based activation approach and its impact on net CRE and for which conditions you associate the largest biases. While detailed case descriptions are not necessary, context should be given for the reader in terms of the conditions of June 2nd-June 5th (for which the bias attribution and sensitivity analysis is done). In particular information with respect to temperature regime, integrated water vapour content, optical depth regime, precipitation characteristics and stability would be useful. Also are these predominantly stratiform or broken cloud-decks? I would also suggest to contextualise your findings in the discussion section in terms of how far you would be comfortable to extrapolate your findings beyond the optically thin (I assume), single-layer cloud regime that you explored here in greater detail.*

A more elaborate description of the prevailing conditions during the period of interest is now given in the revised manuscript. This includes a quantification of the temperature and humidity regime. Additional information is given on the state of the atmospheric boundary layer, as well as on the cloud regime that prevailed during that period.

Regarding the question whether our findings can be extrapolated to conditions beyond the cloud regime on which we focused on in this study (i.e. optically thin single-layer clouds), we additionally analyzed days with multi-layer clouds being present, which was the case in mid June 2017. We find a similar overestimated transmissivity and stronger warming effect of clouds in ICON compared to the observations. This information has been added to the conclusions in the revised manuscript.

*2. You argue the utility and necessity to evaluate and improve kilometre-scale simulations. In this paper you provide a pathway to improve the simulated net CRE for "kilometre-scale" ICON simulations for Arctic low-level clouds, which may even yield to improvements to similar cloud regimes simulated in other regions of the globe. In order to use this approach more widely, it would be helpful to be aware of its potential limitations within the range of "kilometre-scale" grids. Here, you show results of a particular configuration of horizontal (1.2km resolution) and vertical resolution. Terms like "kilometre-scale", "convection-permitting", etc. are often used in the community for a range of resolutions ranging from, say, 1-5 km, which apply all kinds of vertical grid refinement within the boundary layer. How valid do you expect your conclusions to remain across the range of spatial resolutions that fall under the category "kilometre-scale"/"convection-permitting"? Would you expect your TKE fix to droplet activation to work equally well at a (say) 5km grid spacing, or when only half the vertical grid spacing is applied?*

To quantify whether our pathway to improve cloud microphysics in Arctic clouds can also be employed at higher spatial resolution in the horizontal and in the vertical, we did two sensitivity studies. As simulations at 5 km would have implied a substantial effort due to the fact that we would have to completely redo the generation of the input data (i.e. grid, external parameters and forcing data), we only looked at the effects on the outer domain of our set-up, which has a horizontal resolution of 2.4 km. Additionally, we did another simulation at 2.5 km in which we reduced the number of vertical levels from 75 to 50, which is comparable to the vertical resolution of present-day climate models.

The histograms of hydrometeor diameter, number concentration and liquid water content at 2.4 km resolution with 75 vertical levels (Figure 1) and at 2.4 km resolution with 50 vertical levels (Figure 2) are almost identical to the Fig. 6 in the revised manuscript at 1.2 km. Still, 2.5 km is still on the finer end of kilometer-scale simulation.

In our set-up, TKE is used to include subgrid-scale vertical motion in the activation of CCN into cloud droplets. A similar parameterization for the activation of CCN due to subgrid-scale vertical motion has been used at even coarser resolutions of 20 km (Morrison and Pinto, 2005). Nevertheless, in such an activation parameterization, it is crucial that grid-scale TKE is correctly parameterized. In ICON, this quantity is calculated from a prognostic TKE scheme following Raschendorfer (2001). As this TKE scheme is also used for the operational ICON performed at global scale with a resolution of more than 10 km makes us confident that it should also perform reasonably well at resolution larger than 2.4 km. Therefore, we are confident that this activation parameterization can be employed for coarser resolution in ICON and also for kilometer scale simulation in models that employ a two-moment scheme. A summary of these new results has been added to the revised manuscript

[Figure]

Figure 1: As Fig. 6 in the revised manuscript but at a horizontal resolution of 2.4 km and with 75 vertical levels.

[Figure]

Figure 2: As Fig. 6 in the revised manuscript but at a horizontal resolution of 2.4 km and with 50 vertical levels.

*3. In section 3 during your evaluation of surface radiative quantities you argue that you can compensate for the temporal irregularity of your model output (every 3h) by the increased spatial coverage and thus increased sampling of spatial variability. This essentially assumes that spatial and temporal variability are equivalent. This is assumption is commonly made during simulation-observation comparisons. Can you demonstrate this to be valid though forradiative quantities subject to a diurnal cycle?*

The output frequency of our model simulations is 30 minutes (3 hours is the frequency of the forcing

data from the IFS). This implies that the largest temporal difference between an observational data point and the output timestep of ICON is ± 15 minutes. To illustrate the effect of this temporal inconsistency, we plotted the bias in incoming solar radiation at TOA introduced by the limited model output frequency and the applied temporal sampling (Figure 3). The bias is largest ($\pm 14\,\mathrm{W\,m^{-2}}$) at 7 and 19 UTC when the temporal derivative of incoming solar radiation is the largest. During noon when most of the research flights took place, this bias is substantially smaller. Considering that we focused our analysis mainly on cloudy conditions, this maximum bias is further reduced and probably on the order of a few $\mathrm{W\,m^{-2}}$. Additionally, if long enough periods are considered, any bias will eventually average out. Especially for the period of the sensitivity study where only a limited amount of low-level section are available, this can not be fully ensured. Nevertheless, we are confident that this issue will not significantly influence the overall findings in this study as the biases found are almost one order of magnitude larger than the biases introduced by the limited model output frequency. We summarize this result in the revised manuscript.

[Figure]

Figure 3: Bias in incoming solar radiation at TOA at 80° N for 1 June introduced by the limited model output frequency and applied temporal sampling.

*4. Your analysis of biases regarding net CRE is focused on the period of 2-5th of June. In L234 you state that you select this period because you largely are dealing with single-layered low-level clouds and have a high density of flights. I am assuming that the bias in CRE (Fig. 4) is also largest during this time period and for this particular cloud regime as well? Given the significance of the analysis that follows for the overall manuscript, I would include a couple more sentences on this selection for clarity.*

Those days were mainly selected due to similar meteorological conditions that enabled a statistical aggregation of those days. Furthermore, in-situ observations of microphysical properties were performed on all flight days. The day with the largest bias in CRE has been observed on 14 June, but this was a day with a lot of multi-layer clouds present, which made the interpretation of the bias in CRE much harder than for single layer clouds. In the revised manuscript, we give more information of why this period was chosen and also to which extent the bias in CRE can be observed for other meteorological conditions (see also reply to general comment #1).

*5. I agree with your general sentiment conveyed in the introduction and conclusion sections of this manuscript that high-resolution LES simulations are quite limited in their spatial and temporal coverage and that coarser-resolution simulations allow longer-term evaluations over larger domains. Yet I wonder, if you are not subject to the same limitations in this particular application, since you are restricting this evaluation to the location of 15 linear flight tracks within a particular region (although admittedly you can afford to simulate more flight hours), and most of your analysis is focused on the period June 2nd-5th. The argumentation of the benefits and limitations of kilometre-scale versus LES simulations does not seem an essential part of your analysis. Thus I would consider to reduce the emphasis on this point, as this is not something you actually show.*

Indeed, for the limited domain where research flights took place, the larger spatial coverage of simulations at kilometer-scale might not be needed. Nevertheless, the ability of being able to afford a large amount of sensitivity studies would be extremely resource intensive at finer resolution and, therefore, simulations at kilometer-scale are a good compromise. As proposed by the reviewer, we shortened the pros-and-cons discussion of kilometer-scale versus LES simulations in the revised manuscript.

*6. Fig. 6 very clearly shows the bias in simulated cloud properties that are consistent with an overestimation in cloud transmissivity. From the observations you are under constrained and cannot (I presume) say with certainty whether this is a source or sink issue. In your analysis you show that the bias can be fixed by increasing the source in Nd. Can you provide an equally strong argument, that you could not obtain the same improvement, by adjusting the sink? I think this could be done in the context of a discussion of cloud-base or surface precipitation rates, or a couple of additional numerical experiments where you explicitly show that adjustments to the autoconversion rate by: either turning it off altogether – essentially shutting off warm rain – or reducing its efficiency, does not yield the same kind of improvement.*

We performed an additional sensitivity study in which autoconversion was turned off completely. While the effect on liquid water content is comparable to the revised, but not yet scaled CCN activation (see Fig. 7), the cloud droplet number concentration is still slightly underestimated. Furthermore, the shape of particle number size distribution still does not really match the shape of the observed size distribution. Since the CCN profile used in the activation of CCN into cloud droplets within the cloud microphysical scheme is not suited for an Arctic domain as it overestimates the availability of CCN, the underestimated amount of cloud droplets in the simulations with autoconversion turned off is a further indication that it is rather a source than a sink problem. We furthermore looked at CRE for turned off autoconversion (not shown). The effect of turning off autoconversion altogether is comparable to the effect of revised CCN activation (see Fig. 9b), as the CDNC used in the radiation routine is a constant profile and not coupled to the cloud microphysics. As the source-or-sink discussion is an important aspect, we added this discussion to the revised manuscript.

[Figure]

Figure 4: As Figure 6 in the revised manuscript, but with autoconversion turned off.

**General comments**

*1. L44: This seems like a somewhat random selection of LES studies in the Arctic and by no means complete. I suggest to either include a comprehensive list of references, or to make it clear that this list of studies is merely exemplary.*
An "e.g." has been added to clarify that this list of studies is merely exemplary.

*2. L48ff: In addition to the representation of in-cloud turbulence and cloud-top inhomogeneity, LES setup also allows the study and evaluation of microphysical processes (e.g. Ovchinikov et al. (2014), Solomon et al. (2015), Fridlind et al. (2017)) and aerosol-cloud interactions (e.g. Possner et al. (2017), Solomon et al. (2018), Eirund et al. (2019)) at scales where the dynamics and thermodynamics are largely resolved. Since you identify the representation of CCN and the activation process itself as one of your primary sources of bias regarding net CRE. It seems fair to mention this here. Refs:*
*1.Ovchinikov et al (2014): doi:10.1002/2013MS000282 (JAMES)*
*2.Fridlind et al (2017): doi:10.1029/2007JD008646 (JGR)*
*3.Solomon et al (2015): doi:10.5194/acp-15-10631-2015 (ACP)*
*4.Possner et al (2017): doi:10.1002/2016GL071358 (GRL)*
*5.Solomon et al (2018): https://doi.org/10.5194/acp-2018-714 (ACP)*
*6.Eirund et al (2019): https://doi.org/10.5194/acp-19-9847-2019 (ACP)*

Thank you for pointing us to these references. We added a sentence that highlights the use of LES with regards to the evaluation of cloud microphysical processes and aerosol-cloud interactions in the Arctic.

*3. L102: What is your reasoning for using the all or nothing cloud-cover scheme? Did it impact your results?*
The all-or-nothing cloud cover scheme was mainly chosen to facilitate the comparison of the simulations with the observations. Having fractional cloud cover in the simulation would imply the need to divide microphysical properties by the fractional cloud cover to get the respective in-cloud values, that are present in the observational dataset. For that reason, we decided to use an all-or-nothing cloud cover scheme where this is not necessary. At the resolutions used in this study, an all-or-nothing cloud cover scheme might miss some clouds as the necessary saturation humidity might not be reached, which might be especially problematic for weak dynamical forcing. The cloud fields and also the radiative properties of clouds between the all-or-nothing cloud cover scheme and a cloud cover scheme that allows for fractional cloud cover were relatively similar along the flight track of the research flights, which made us confident that resolving clouds at grid scale only is sufficient for our set-up.

*4. L132: "The daily averaged observed albedo is parameterized as a function of day of the year". I did not follow this. Did you not simply prescribe the daily mean albedo value from the full sea-ice covered surface observations. So how is it a "function" of the day of year?*
Due to the fact that the campaigns took place at the onset of the melting period, the sea ice albedo significantly reduced in that timespan. To this end, we prescribed the sea ice albedo derived from aircraft observations over fully sea ice covered regions to be consistent with that evolution and, therefore, have parameterized the sea ice albedo as a function of time (i.e. day of the year). The description of this approach has been revised to be better understandable.

*5. L177/178: Why did you not fix the sea ice fraction in a similar fashion as the sea ice albedo in your simulation setup to exclude the impact of biases from essentially prescribed surface properties?*
We opted against prescribing sea ice fraction because one would only be able to prescribe sea ice fraction along the flight track as a generalized formulation would not be possible due to the highly spatially variable nature of sea ice fraction. Such a spot change would not significantly affect the thermodynamic profile in a dynamic clouds field and, therefore, not affect the resulting consequences on cloud macro- and microphysical properties. For that reason, only the differences in surface albedo

will affect the radiative effect of clouds along the flight track, which we qualitatively discuss in the manuscript.

*6. L217: I agree with your conclusion that the underestimated cooling in the solar spectral range is likely due to an incorrect simulation of cloud transmissivity, rather than remaining biases in surface albedo. As this a central aspect to your overall argument, I was wondering if you could not show this explicitly. Do your conclusions remain the same if you restrict the phase space your analysis of the observations to surface albedo values $< 0.8$ such as to match the simulations?*

Constraining the phase space of the observations by imposing an upper limit for the allowed surface albedo values is indeed a good idea. Instead of the proposed threshold of 0.8, we decided to choose an upper bound of 0.85, which is equivalent to the daily averaged maximum albedo value used in our adapted albedo parameterization. This threshold has now been applied to all plots in the revised manuscript. The effect of this upper albedo threshold can mainly be seen in changes of the radiative properties in the ACLOUD data due to the reduced surface reflectivity. Due to the fact that the respective datapoints in both datasets have to fulfill the chosen condition at the same time, also small changes can be observed for the ICON data. Nevertheless, the general conclusions using an upper threshold for the surface albedo stay the same.

*7. L231: I personally would argue that cloud water content is to first order a thermodynamic variable and thus also a macrophysical variable that is adjusted by microphysical processes (i.e. the efficiency of autoconversion/accretion in warm-phase clouds anyway). Especially in a model with saturation adjustment I have a hard time referring to qc as purely microphysical, but can be convinced.*

It is correct that cloud water content should not solely be considered as a microphysical variable. We clarified that in the revised manuscript.

*8. "it shows a slight underestimation". Can you quantify this? What is the average/median cloud depth?*

The mean cloud depth bias of the model compared to the observation is 65 m. This quantification has been added to the revised manuscript.

*9. L260: I would suggest to be more specific/quantitative here, as this is a key argument in your assertion that the bias stems predominantly from biases in cloud water content and droplet concentration. For the typical cloud optical depth seen in your simulations or during the observation record, how large would a geometrical cloud depth bias have to be to affect transmissivity substantially? How does that quantity relate to your biases assessed? In that context, I am not sure Fig. 5 is best suited. I wonder if a PDF-based comparison is not more informative. Cloud transmissivity is strongly non-linearly related to geometrical cloud depth. Thus biases in the distribution of geometrical depth (although means may agree), could induce substantial biases in mean cloud transmissivity. To follow your line of assertion, the argument that cloud depth biases are unlikely to contribute signficantly should be strengthened quantitatively.*

To explore the effect of a larger geometrical cloud depth on the CRE, we used in-situ profiles of LWC from 4 June that have been observed close to R/V Polarstern and linearly interpolated the LWC with altitude (Figure 5). To calculate the CRE, we used offline radiative transfer simulations (for more details on those simulations, see section 3.2 in the revised manuscript). The albedo used in these simulations is set 0.835, which is the mean albedo value for that day. From the profile of LWC and temperature, we estimate the geometrical cloud depth to be around 290 m for that day, which is in accordance with what has been observed from R/V Polarstern. Looking at the change of $CRE_{sol}$ with vertical cloud extend, we find an almost linear relationship between the two quantities. If one would reduce the vertical cloud extend by 65 m, the $CRE_{sol}$ would approximately increase by $5\,\mathrm{W\,m^{-2}}$. We repeated the estimation of the bias introduced by the non-matching vertical cloud extend for other vertical profiles that had lower adiabaticity factors than in this case and the obtained biases in $CRE_{sol}$ were in a similar range. As the bias in $CRE_{sol}$ between ICON and the observations for the days of the sensitivity studies is more than $20\,\mathrm{W\,m^{-2}}$, we are confident that the bulk of this bias is actually

caused by misrepresented cloud microphysical properties in ICON. The deviations to the solar CRE observed by ACLOUD for the period from 2 June to 5 June (see supplement) can be explained by the higher albedo used in the offline radiative transfer simulations compared to the flight sections used there. This is now reported in the revised manuscript.

[Figure]

Figure 5: (a) Observed in-situ vertical profile of temperature (red) and liquid water content (blue) for vertical profile near R/V Polarstern on 4 June. The black line is the linearly interpolated LWC with an adiabaticity factor $f_{\mathrm{ad}}$ of 0.98.(b) LWP as a function of geometrical cloud depth using with $f_{\mathrm{ad}}$ of 0.98. The dashed lines indicate the observed geometrical cloud depth and LWP. (c) $CRE_{\mathrm{sol}}$ as a function of geometrical cloud depth. $CRE_{\mathrm{sol}}$ has been derived from offline radiative transfer simulations using the respective LWP as calculated in (b). Deviation of $CRE_{\mathrm{sol}}$ compared to what has been observed by ACLOUD (see Fig. S3 in the supplement) stem from lower albedo ($\sim 0.79$) in these flight sections compared to the offline radiative transfer simulations (0.835).

*10. L270: "droplets plus ice crystals": essentially droplet concentration at $<1\,cm^{-3}$. I would not expect to see any impact of ice in the shown range. It may be worth to state explicitly.*
Done.

*11. L300: Can you provide a quantitative estimate of rain rate? Do you have any constraint here from the observation?*
We added a quantitative estimate of modeled rain- and snow rate in the region around R/V Polarstern to the revised manuscript. Unfortunately, no observations of surface precipitation was available.

*12. Fig 2: In the range of surface albedos between 0.6-0.8 where the number of occurrence is highest, the simulations show a much considerably narrower range of $F_{net,sol}$ than the observations. Do you have any idea whether this is indicative of a model bias, or simply a sampling issue between the simulation and observation datasets?*
As stated in the manuscript, low albedo values are related to days towards the end of the campaign when cloud free conditions were present. Such a day was the 25 June, that a had a relatively large amount of low-level sections. In contrast to the observations, clouds were present in the model on that day, causing a negative bias in $F_{net,sol}$ of more than $80\,\mathrm{W\,m^2}$ between the model an the observations. Due to relatively large amount of observations being present for that day, this effect of this day can also be seen in Figure 2. Therefore, this narrower range of $F_{net,sol}$ can be considered to be both, a model and a sampling bias. It is exactly for that reason why we filtered both datasets so that they are in the same radiative state.

*13. Fig 5: I personally find it hard to draw quantitative conclusions from this plot going beyond the overall range of values in observations and simulations. You can sort of see that geometrical cloud*

*depth is likely underestimated, but its hard to tell due to the many overlapping points where the real density of points is. As suggested previously, I wonder if a PDF comparison would not be more informative*

Depicting the bias in geometrical cloud depth in a histogram is indeed a good idea and we revised Fig. 5 accordingly.

*14. Fig. 6-8: Panel a): I am fairly sure the ICON and ACLOUD lines are swapped? Otherwise there would be a mismatch between your figure and the discussion and the results of sections 4.2ff. Panel b): The ACLOUD in Fig6 is slightly different to 7/8. Why?*

Thanks for spotting that issue, the lines in Fig. 6-8: Panel a) are indeed swapped. This has been corrected in the revised manuscript. As discussed for radiative properties, the cloud field is different between the different sensitivity studies. As only datapoints are being used when both, the model and the observation, are within a cloud at the same time, the histograms do slightly differ. This is clarified in the revised manuscript.

*15. Fig. 9: Ultimately the net cloud-radiative effect is of interest, but your argument primarily relates to CREsol. What is the impact of CREsol alone?*

The CRE is mainly mediated by its solar component in all the sensitivity studies. The terrestrial components are in good agreement with the observationally derived terrestrial CRE components. We included this information to the revised manuscript and included the respective figures in a supplement to the article.

*16. Table 1: I personally would find a a total number of included flight hours as part of the caption helpful.*

A total of approx. 116 flight hours has been used for this comparison. We added this information to the caption.

**Edits**

*1. L165: suggested rephrase "to the previous comparison" to "to the previously used model setup".*
*2. L248: "the the"*
*3. L291: suggest rephrase "than observed the mean of" to "than the observed mean of"*
*4. L423: Typo? Underestimation of geometrical cloud depth, right?*
*5. L424: "represent" instead of "simulate" (since you do not really simulate it)*

All remarks have been implemented as proposed.

**Further revision**

In line 186 of the submitted manuscript, the threshold for a surface to be classified as sea ice covered should be 0.7, not 0.5. This has been corrected in the revised manuscript.

**References**

Morrison, H. and Pinto, J. O.: Mesoscale Modeling of Springtime Arctic Mixed-Phase Stratiform Clouds Using a New Two-Moment Bulk Microphysics Scheme, Journal of the Atmospheric Sciences, 62, 3683–3704, https://doi.org/10.1175/JAS3564.1, 2005.

Raschendorfer, M.: The new turbulence parameterization of LM, COSMO newsletter No. 1, pp. 89–97, 2001.

---

## Author Comment (AC2) · 18 Sep 2020

**Response to referee comment #2**

*In this paper, the authors compare simulations using the ICON model to observationsfrom the ACLOUD and PASCAL campaigns. They find that the ICON simulations predict a more strongly positive cloud radiative effect (CRE) than that derived from the ACLOUD observations. They then determine that an important contribution to this difference is the small number of cloud condensation nuclei (CCN) activated in the ICON model, which subsequently results in low cloud liquid water contents. They improve the model results by accounting for the effects of subgrid-scale turbulence on cloud droplet activation and by scaling their assumed CCN profile. I feel that the study merits publication, provided that the following comments are addressed.*

We thank the reviewer for the constructive comments that helped to improve the manuscript.

**General comments**

*1. The authors briefly mention cloud ice in a few places in the paper, but they largely restrict their analysis to liquid cloud water. Some definitive or quantified statements about the contributions of ice clouds to the radiation balance or hydrometeor concentrations, both in ICON and in the observations, would be welcome. Could differences in the amount of frozen cloud make a significant contribution to differences in the surface radiation balance or the cloud radiative effect between the model and the observations?*

From the observational side, it is difficult the quantify contribution of ice clouds to the radiation balance or hydrometeor concentrations as the amount of ice in the clouds during ACLOUD and especially during the period of our sensitivity study was relatively low and often times below the detection threshold of the in-situ probes. Looking at the ICON model, we have performed a sensitivity analysis in which we turned off any radiative effect of cloud ice. If one compares the radiative variables like surface CRE (see Figure 1 and Figure 2) and $F_{net}$ at the surface (not shown), the differences between our basic set up of ICON and the one without an effect of cloud ice on the radiative field is small and on the order of 1 $\mathrm{W\,m^{-2}}$. This is due to the already low cloud ice fraction in the model, which also causes the radiative effect of cloud ice to be low. Due to the limitations of the observational dataset in terms of cloud ice, it is hard to constrain the model from the observational side. Therefore, any estimation of the impact of cloud ice on the radiative balance has to be interpreted with some caution. We added this information to the revised manuscript.

[Figure]

Figure 1: As Fig. 4 in the revised manuscript but for the period from 2 June to 5 June.

*2. Sect. 3.2, p9: The authors mention here that the CRE is calculated from the observations through the methods of Stapf et al. (2019a). Given that there are potential inconsistencies in the calculated CRE between the model and the observations, just a little more detail on the radiative transfer simulations of Stapf et al. (2019a) seems prudent here.*

In the revised manuscript, more information on the radiative transfer simulations are given.

[Figure]

Figure 2: As Fig. 4 in the revised manuscript but for the period from 2 June to 5 June and without effect of cloud ice on radiation.

*The authors mention that "While the prescribed functional dependence of the sea ice albedo has been derived for cloudless and cloudy conditions, the surface albedo that is used to derive the CRE from the observations is for cloudy-sky only. This can lead to inconsistencies between the modeled and observed CRE (Stapf et al., 2019a)." However, If I understand correctly, the radiative transfer simulations of Stapf et al. (2019a) account for cloud surface-albedo interactions. Given that the surface albedo is prescribed in the ICON simulations, these cloud-surface-albedo interactions will not be accounted for in the ICON simulations. Therefore, wouldn't it be a more consistent comparison if the cloud-surface-albedo interactions were also neglected in CRE calculations based on the observed data? Can the authors comment on this?*

The radiative transfer simulations to derive the CRE from the observation are different from the ones in Stapf et al. (2019a) as in our study, the albedo from all-sky conditions was used. All-sky albedo was also used to derive the functional dependency used that we implemented into ICON for the purpose of this study. We now explicitly state that all-sky albedo was used and removed a misleading citation to Stapf et al. (2019a) to avoid confusion.

**Specific comments and technical corrections**

*p2, line 38: optical → optically*
Changed.

*p2, lines 44-47: Please improve the clarity of this sentence.*
Following the advise by reviewer #1 to reduce the LES vs kilometer-scale simulation, this sentences has been removed in the revised manuscirpt.

*p3, line 74: sea ice covered → sea-ice-covered*
Changed.

*p3, line 83: unmatched parenthesis: "given (for"*
Parenthesis added.

*p3, line 84: refer → refer the reader to*
Changed.

*p5, line 108: "general feature of ICON." Perhaps the authors mean "generally representative of ICON"?*

Changed.

*p5, line 120: "caused by the way how our simulations" Please either choose "the way that" or "how".*
Changed to "how".

*p8, line 176 "sea ice covered surface". This should be either "the sea-ice-covered surface" or "sea-ice-covered surfaces".*
Changed to "sea-ice-covered surfaces".

*p8, line 189: "Figure 3 a" → "Figure 3a"*
We refer to Figure 3a in the following sentence, and this sentence was intended to generally introduce this figure.

*p9, line 200: Please insert a comma after "without clouds"*
Comma inserted.

*p9, line 202: "measurements of atmospheric/surface observations". Perhaps the authors mean "atmospheric or surface measurements" or "atmospheric or surface observations"?*
Here, we refer to observatoins of the atmosphere (i.e. dropsonds) and of surface properties (i.e. albedo). We reformulated this sentence to be more concise.

*p9, line 211: Please either choose "The way that" or "How".*
Changed to "The way that".

*p9, line 212: "allows to narrow down, which effect" → either "allows us to narrow down which effect" or "allows one to narrow down which effect".*
Changed to "allows us to narrow down which effect".

*p9, line 212: "If clouds would be" → "If clouds were"*
Changed.

*p9, line 215: "fraction" → "ratio"*
Changed.

*p10, line 226: "which allows to" → "which allows us to"*
Changed.

*p11, line 260: "extend" → "extent"*
Changed.

*p13, line 301: large → larger*
Changed.

*p13, line 302: stems → stem*
Changed.

*p14, lines 327-328:The overestimation of small hydrometeors mentioned here seems to be in contradiction to the statements of p12, lines 278-280.*
Here, we refer to the overestimation of small hydrometeors in Schemann and Ebell (2020). Due to the much finer resolution of their ICON simulations, the activation of CCN into cloud droplets can be sufficiently resolved and any bias is only to the unsuited background CCN profile for an Arctic domain. Neverthless, we revised this sentence to make that clearer that we refer to the simulations in Schemann and Ebell (2020).

*p16, lines 393-394: Since the last simulation discussed was not the default set-up but instead was the one using the revised CCN activation scheme, most readers would assume that the authors are comparing the simulation with the CCN scaled by 0.4 to the revised CCN activation simulation. The authors need to make it clear that they are comparing this simulation to the default set-up.*

It has been clarified in the revised manuscript that we scaled the revised CCN activation simulation and not the default set-up.

*p17, lines 411 and 414: Do the authors mean Figure 9f instead of 9e?*

Yes indeed, Figure 9f is the one we refer to. This has be changed accordingly.

*p20, eq. B3: If I divide eq. B2 with $k = 3$ by eq. B2 with $k = 2$, I find the trailing factor to be $A^{-1/\mu}$, not $\lambda^{-1/\mu}$. Is the error in eq. B2 or eq. B3?*

Thanks for thoroughly going through the equations. Indeed, there is a typo in B2 as there has to be a $\lambda$ in the denominator instead of $A$. This has been corrected in the revised manuscript.

*Figure 1 caption: "inner domain has a" → "inner domain (red) has a"*

Changed as proposed.

*Figure 5 and p11, lines 258-261: There is significant overlap in the points on this plot, which makes it difficult to tell, for example, how large a fraction of the data have observed cloud depth < 0.4 and modelled cloud depth < 0.2. This also means that it is difficult to judge the degree of underestimation of the cloud depths. I don't have a perfect solution for this issue, but the authors may wish to consider making the data points partially transparent, or substitution of the scatter plot with a histogram (with different subplots for the different observation days, if the authors wish). I am open to other solutions, or arguments from the authors in favour of the current plot. In any case, the median values of the modelled and observed cloud depths should be provided to help the reader quantify the degree of underprediction. The means and standard deviations may also be helpful.*

We revised this figure and now display the bias in the form of a histogram. The mean and the standard deviation of the depicted histogram are given in the revised manuscript.

*Figure 6, Figure 7, and Figure 8: The red lines for panels b and c are very similar in the three figures, but not quite identical. Note for instance that the peak in frequency of hydrometeor number concentration is > 100 in Figure 6 and < 100 in Figure 7 and Figure 8. Rather than state in the captions that the lines are identical, the authors instead should very briefly remind the reader why the lines differ slightly. Also, it seems that the red and blue lines are reversed in panel a in all three figures.*

The lines in Fig. 6-8: Panel a) are indeed swapped, which has been corrected in the revised manuscript.

*Figure 9: It would be prudent to remind the reader in the caption that the red lines differ slightly due to the sampling that is applied.*

We added a remark in the caption of Figs. 6-9 that the red lines differ due to the sampling strategy employed.

**Further revision**

In line 186 of the submitted manuscript, the threshold for a surface to be classified as sea ice covered should be 0.7, not 0.5. This has been corrected in the revised manuscript.

**References**

Schemann, V. and Ebell, K.: Simulation of mixed-phase clouds with the ICON large-eddy model in the complex Arctic environment around Ny-Ålesund, Atmospheric Chemistry and Physics, 20, 475–485, https://doi.org/10.5194/acp-20-475-2020, 2020.